# Phase Retrieval Under a Generative Prior

**Paul Hand**[*]
Northeastern University
p.hand@northeastern.edu

**Oscar Leong**
Rice University
oscar.f.leong@rice.edu

**Vladislav Voroninski**
Helm.ai
vlad@helm.ai

## Abstract

We introduce a novel deep learning inspired formulation of the *phase retrieval problem*, which asks to recover a signal $y_0 \in \mathbb{R}^n$ from $m$ quadratic observations, under structural assumptions on the underlying signal. As is common in many imaging problems, previous methodologies have considered natural signals as being sparse with respect to a known basis, resulting in the decision to enforce a generic sparsity prior. However, these methods for phase retrieval have encountered possibly fundamental limitations, as no computationally efficient algorithm for sparse phase retrieval has been proven to succeed with fewer than $O(k^2 \log n)$ generic measurements, which is larger than the theoretical optimum of $O(k \log n)$. In this paper, we propose a new framework for phase retrieval by modeling natural signals as being in the range of a deep generative neural network $G : \mathbb{R}^k \to \mathbb{R}^n$. We introduce an empirical risk formulation that has favorable global geometry for gradient methods, as soon as $m = O(kd^2 \log n)$, under the model of a $d$-layer fully-connected neural network with random weights. Specifically, when suitable deterministic conditions on the generator and measurement matrix are met, we construct a descent direction for any point outside of a small neighborhood around the true $k$-dimensional latent code and a negative multiple thereof. This formulation for structured phase retrieval thus benefits from two effects: generative priors can more tightly represent natural signals than sparsity priors, and this empirical risk formulation can exploit those generative priors at an information theoretically optimal sample complexity, unlike for a sparsity prior. We corroborate these results with experiments showing that exploiting generative models in phase retrieval tasks outperforms both sparse and general phase retrieval methods.

## 1 Introduction

We study the problem of recovering a signal $y_0 \in \mathbb{R}^n$ given $m \ll n$ phaseless observations of the form $b = |Ay_0|$ where the measurement matrix $A \in \mathbb{R}^{m \times n}$ is known and $|\cdot|$ is understood to act entrywise. This is known as the *phase retrieval problem*. In this work, we assume, as a prior, that the signal $y_0$ is in the range of a generative model $G : \mathbb{R}^k \to \mathbb{R}^n$ so that $y_0 = G(x_0)$ for some $x_0 \in \mathbb{R}^k$. To recover $y_0$, we first recover the original latent code $x_0$ corresponding to it, from which $y_0$ is obtained by applying $G$. Hence we study the *phase retrieval problem under a generative prior* which asks:

$$\text{find } x \in \mathbb{R}^k \text{ such that } b = |AG(x)|.$$

We will refer to this formulation as Deep Phase Retrieval (DPR). The phase retrieval problem has applications in X-ray crystallography [21, 29], optics [34], astronomical imaging [14], diffraction

---

[*]Authors are listed in alphabetical order.

imaging [5], and microscopy [28]. In these problems, the phase information of an object is lost due to physical limitations of scientific instruments. In crystallography, the linear measurements in practice are typically Fourier modes because they are the far field limit of a diffraction pattern created by emitting a quasi-monochromatic wave on the object of interest.

In many applications, the signals to be recovered are compressible or sparse with respect to a certain basis (e.g. wavelets). Many researchers have attempted to leverage sparsity priors in phase retrieval to yield more efficient recovery algorithms. However, these methods have been met with potentially severe fundamental limitations. In the Gaussian measurement regime where $A$ has i.i.d. Gaussian entries, one would hope that recovery of a $k$-sparse $n$-dimensional signal is possible with $O(k \log n)$ measurements. However, there is no known method to succeed with fewer than $O(k^2 \log n)$ measurements. Moreover, [26] proved that the semidefinite program PhaseLift cannot outperform this suboptimal sample complexity by direct $\ell_1$ penalization. This is in stark contrast to the success of leveraging sparsity in linear compressed sensing to yield optimal sample complexity. Hence enforcing sparsity as a generic prior in phase retrieval may be fundamentally limiting sample complexity.

**Our contribution.**   We show information theoretically optimal sample complexity[2] for structured phase retrieval under generic measurements and a novel nonlinear formulation based on empirical risk and a generative prior. In this work, we suppose that the signal of interest is the output of a generative model. In particular, the generative model is a $d$-layer, fully-connected, feed forward neural network with Rectifying Linear Unit (ReLU) activation functions and no bias terms. Let $W_i \in \mathbb{R}^{n_i \times n_{i-1}}$ denote the weights in the $i$-th layer of our network for $i = 1, \ldots, d$ where $k = n_0 < n_1 < \cdots < n_d$. Given an input $x \in \mathbb{R}^k$, the output of the the generative model $G : \mathbb{R}^k \to \mathbb{R}^{n_d}$ can be expressed as

$$G(x) := \mathrm{relu}\left(W_d \ldots \mathrm{relu}(W_2(\mathrm{relu}(W_1 x))) \ldots \right)$$

where $\mathrm{relu}(x) = \max(x, 0)$ acts entrywise. We further assume that the measurement matrix $A$ and each weight matrix $W_i$ have i.i.d. Gaussian entries. The Gaussian assumption of the weight matrices is supported by empirical evidence showing neural networks, learned from data, that have weights that obey statistics similar to Gaussians [1]. Furthermore, there has also been work done in establishing a relationship between deep networks and Gaussian processes [25]. Nevertheless, we will introduce deterministic conditions on the weights for which our results hold, allowing the use of other distributions.

To recover $x_0$, we study the following $\ell_2$ empirical risk minimization problem:

$$\min_{x \in \mathbb{R}^k} f(x) := \frac{1}{2}\Big\| |AG(x)| - |AG(x_0)| \Big\|^2. \tag{1}$$

Due to the non-convexity of the objective function, there is no a priori guarantee that gradient descent schemes can solve (1) as many local minima may exist. In spite of this, our main result illustrates that the objective function exhibits favorable geometry for gradient methods. Moreover, our result holds with information theoretically optimal sample complexity:

**Theorem 1** (Informal). *If we have a sufficient number of measurements $m = \Omega(kd \log(n_1 \ldots n_d))$ and our network is sufficiently expansive at each layer $n_i = \Omega(n_{i-1} \log n_{i-1})$, then there exists a descent direction $v_{x,x_0} \in \mathbb{R}^k$ for any non-zero $x \in \mathbb{R}^k$ outside of two small neighborhoods centered at the true solution $x_0$ and a negative multiple $-\rho_d x_0$ with high probability. In addition, the origin is a local maximum of $f$. Here $\rho_d > 0$ depends on the number of layers $d$ and $\rho_d \to 1$ as $d \to \infty$.*

Our main result asserts that the objective function does not have any spurious local minima away from neighborhoods of the true solution and a negative multiple of it. Hence if one were to solve (1) via gradient descent and the algorithm converged, the final iterate would be close to the true solution or a negative multiple thereof. The proof of this result is a concentration argument. We first prove the sufficiency of two deterministic conditions on the weights $W_i$ and measurement matrix $A$. We then show that Gaussian $W_i$ and $A$ satisfy these conditions with high probability. Finally, using these two conditions, we argue that the specified descent direction $v_{x,x_0}$ concentrates around a vector $h_{x,x_0}$ that is continuous for non-zero $x \in \mathbb{R}^k$ and vanishes only when $x \approx x_0$ or $x \approx -\rho_d x_0$.

Rather than working against potentially fundamental limitations of polynomial time algorithms, we examine more sophisticated priors using generative models. Our results illustrate that these priors are,

in reality, less limiting in terms of sample complexity, both by providing more compressibility and by being able to be more tightly enforced.

**Prior methodologies for general phase retrieval.** In the Gaussian measurement regime, most of the techniques to solve phase retrieval problems can be classified as convex or non-convex methods. In terms of convex techniques, lifting-based methods transform the signal recovery problem into a rank-one matrix recovery problem by *lifting* the signal into the space of positive semidefinite matrices. These semidefinite programming (SDP) approaches, such as PhaseLift [9], can provably recover any $n$-dimensional signal with $O(n \log n)$ measurements. A refinement on this analysis by [7] for PhaseLift showed that recovery is in fact possible with $O(n)$ measurements. Other convex methods include PhaseCut [33], an SDP approach, and linear programming algorithms such as PhaseMax, which has been shown to achieve $O(n)$ sample complexity [17].

Non-convex methods encompass alternating minimization approaches such as the original Gerchberg-Saxton [16] and Fienup [15] algorithms and direct optimization algorithms such as Wirtinger Flow [8]. These latter methods directly tackle the least squares objective function

$$\min_{y \in \mathbb{R}^n} \frac{1}{2} \left\| |Ay|^2 - |Ay_0|^2 \right\|^2. \tag{2}$$

In the seminal work, [8] show that through an initialization via the spectral method, a gradient descent scheme can solve (2) where the gradient is understood in the sense of Wirtinger calculus with $O(n \log n)$ measurements. Expanding on this, a later study on the minimization of (2) in [31] showed that with $O(n \log^3 n)$ measurements, the energy landscape of the objective function exhibited global benign geometry which would allow it to be solved efficiently by gradient descent schemes without special initialization. There also exist amplitude flow methods that solve the following non-smooth variation of (2):

$$\min_{y \in \mathbb{R}^n} \frac{1}{2} \left\| |Ay| - |Ay_0| \right\|^2. \tag{3}$$

These methods have found success with $O(n)$ measurements [13] and have been shown to empirically perform better than intensity-based methods using the squared formulation in (2) [37].

**Sparse phase retrieval.** Many of the successful methodologies for general phase retrieval have been adapted to try to solve sparse phase retrieval problems. In terms of non-convex optimization, Wirtinger Flow type methods such as Thresholded Wirtinger Flow [6] create a sparse initializer via the spectral method and perform thresholded gradient descent updates to generate sparse iterates to solve (2). Another non-convex method, SPARTA [35], estimates the support of the signal for its initialization and performs hard thresholded gradient updates to the amplitude-based objective function (3). Both of these methods require $O(k^2 \log n)$ measurements for a generic $k$-sparse $n$-dimensional signal to succeed, which is more than the theoretical optimum $O(k \log n)$.

While lifting-based methods such as PhaseLift have been proven unable to beat the suboptimal sample complexity $O(k^2 \log n)$, there has been some progress towards breaking this barrier. In [19], the authors show that with an initializer that sufficiently correlates with the true solution, a linear program can recover the sparse signal from $O(k \log \frac{n}{k})$ measurements. However, the best known initialization methods require at least $O(k^2 \log n)$ measurements [6]. Outside of the Gaussian measurement regime, there have been other results showing that if one were able to design their own measurement matrices, then the optimal sample complexity could be reached [22]. For example, [2] showed that assuming the measurement vectors were chosen from an incoherent subspace, then recovery is possible with $O(k \log \frac{n}{k})$ measurements. However, these results would be difficult to generalize to the experimental setting as their design architectures are often unrealistic. Moreover, the Gaussian measurement regime more closely models the experimental Fourier diffraction measurements observed in, for example, X-ray crystallography. As Fourier models are the ultimate goal, results towards lowering this sample complexity in the Gaussian measurement regime must be made or new modes of regularization must be explored in order for phase retrieval to advance.

**Related work.** There has been recent empirical evidence supporting applying a deep learning based approach to holographic imaging, a phase retrieval problem. The authors in [18] show that a neural network with ReLU activation functions can learn to perform holographic image reconstruction. In particular, they show that compared to current approaches, this neural network based method requires

less measurements to succeed and is computationally more efficient, needing only one hologram to reconstruct the necessary images.

Furthermore, there have been a number of recent advancements in leveraging generative priors over sparsity priors in compressed sensing. In [4], the authors considered the least squares objective

$$\min_{x \in \mathbb{R}^k} \frac{1}{2} \left\| AG(x) - AG(x_0) \right\|^2. \tag{4}$$

They provided empirical evidence showing that 5-10X fewer measurements were needed to succeed in recovery compared to standard sparsity-based approaches such as Lasso. In terms of theory, they showed that if $A$ satisfied a restricted eigenvalue condition and if one were able to solve (4), then the solution would be close to optimal. The authors in [20] analyze the same optimization problem in [4] but exhibit global guarantees regarding the non-convex objective function. Under particular conditions about the expansivity of each neural network layer and randomness assumptions on their weights, they show that the energy landscape of the objective function does not have any spurious local minima. Furthermore, there is always a descent direction outside of two small neighborhoods of the global minimum and a negative scalar multiple of it. The success of leveraging generative priors in compressed sensing along with the sample complexity bottlenecks in sparse phase retrieval have influenced this work to consider enforcing a generative prior in phase retrieval to surpass sparse phase retrieval's current theoretical and practical limitations.

**Notation.** Let $(\cdot)^\top$ denote the real transpose. Let $[n] = \{1, \ldots, n\}$. Let $\mathcal{B}(x, r)$ denote the Euclidean ball centered at $x$ with radius $r$. Let $\| \cdot \|$ denote the $\ell_2$ norm for vectors and spectral norm for matrices. For any non-zero $x \in \mathbb{R}^n$, let $\hat{x} = x/\|x\|$. Let $\Pi^1_{i=d} W_i = W_d W_{d-1} \ldots W_1$. Let $I_n$ be the $n \times n$ identity matrix. Let $\mathcal{S}^{k-1}$ denote the unit sphere in $\mathbb{R}^k$. We write $c = \Omega(\delta)$ when $c \geqslant C\delta$ for some positive constant $C$. Similarly, we write $c = O(\delta)$ when $c \leqslant C\delta$ for some positive constant $C$. When we say that a constant depends polynomially on $\epsilon^{-1}$, this means that it is at least $C\epsilon^{-k}$ for some positive $C$ and positive integer $k$. For notational convenience, we write $a = b + O_1(\epsilon)$ if $\|a - b\| \leqslant \epsilon$ where $\| \cdot \|$ denotes $|\cdot|$ for scalars, $\ell_2$ norm for vectors, and spectral norm for matrices. Define $\mathrm{sgn} : \mathbb{R} \to \mathbb{R}$ to be $\mathrm{sgn}(x) = x/|x|$ for non-zero $x \in \mathbb{R}$ and $\mathrm{sgn}(x) = 0$ otherwise. For a vector $v \in \mathbb{R}^n$, $\mathrm{diag}(\mathrm{sgn}(v))$ is $\mathrm{sgn}(v_i)$ in the $i$-th diagonal entry and $\mathrm{diag}(v > 0)$ is 1 in the $i$-th diagonal entry if $v_i > 0$ and 0 otherwise.

## 2 Algorithm

While our main result illustrates that the objective function exhibits favorable geometry for optimization, it does not guarantee recovery of the signal as gradient descent algorithms could, in principle, converge to the negative multiple of our true solution. Hence we propose a gradient descent scheme to recover the desired solution by escaping this region. First, consider Figure 1 which illustrates the behavior of our objective function *in expectation*, i.e. when the number of measurements $m \to \infty$.

We observe two important attributes of the objective function's landscape: (1) there exist two minima, the true solution $x_0$ and a negative multiple $-\beta x_0$ for some $\beta > 0$ and (2) if $z \approx x_0$ while $w \approx -\beta x_0$, we have that $f(z) < f(w)$, i.e. the objective function value is lower near the true solution than near its negative multiple. This is due to the fact that the true solution is in fact the global optimum.

Based on these attributes, we will introduce a gradient descent scheme to converge to the global minimum. First, we define some useful quantities. For any $x \in \mathbb{R}^k$ and matrix $W \in \mathbb{R}^{n \times k}$, define $W_{+,x} := \mathrm{diag}(Wx > 0)W$. That is, $W_{+,x}$ keeps the rows of $W$ that have a positive dot product with $x$ and zeroes out the rows that do not. We will extend the definition of $W_{+,x}$ to each layer of weights $W_i$ in our neural network. For $W_1 \in \mathbb{R}^{n_1 \times k}$ and $x \in \mathbb{R}^k$, define $W_{1,+,x} := \mathrm{diag}(W_1 x > 0)W_1$. For each layer $i \in [d]$, define

$$W_{i,+,x} := \mathrm{diag}(W_i W_{i-1,+,x} \ldots W_{2,+,x} W_{1,+,x} x > 0)W_i.$$

$W_{i,+,x}$ keeps the rows of $W_i$ that are active when the input to the generative model is $x$. Then, for any $x \in \mathbb{R}^k$, the output of our generative model can be written as $G(x) = (\Pi^1_{i=d} W_{i,+,x})x$. For any $z \in \mathbb{R}^n$, define $A_z := \mathrm{diag}(\mathrm{sgn}(Az))A$. Note that $|AG(x)| = A_{G(x)} G(x)$ for any $x \in \mathbb{R}^k$.

Since a gradient descent scheme could in principle be attracted to the negative multiple, we exploit the geometry of the objective function's landscape to escape this region. First, choose a random initial

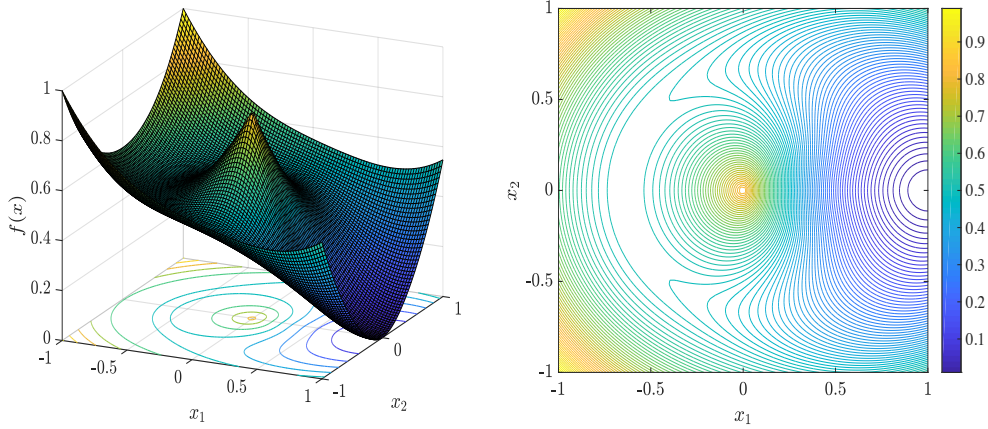

Figure 1: Surface (left) and contour plot (right) of objective function with $m \to \infty$ and true solution $x_0 = [1, \ 0]^\top \in \mathbb{R}^2$.

iterate for gradient descent $x_1 \neq 0$. At each iteration $i = 1, 2, \ldots$, compute the descent direction

$$v_{x_i, x_0} := (\Pi_{i=d}^1 W_{i,+,x_i})^\top A_{G(x_i)}^\top \left( |AG(x_i)| - |AG(x_0)| \right).$$

This is the gradient of our objective function $f$ where $f$ is differentiable. Once computed, we then take a step in the direction of $-v_{x_i, x_0}$. However, prior to taking this step, we compare the objective function value for $x_i$ and its negation $-x_i$. If $f(-x_i) < f(x_i)$, then we set $x_i$ to its negation, compute the descent direction and update the iterate. The intuition for this algorithm relies on the landscape illustrated in Figure 1: since the true solution $x_0$ is the global minimum, the objective function value near $x_0$ is smaller than near $-\rho_d x_0$. Hence if we begin to converge towards $-\rho_d x_0$, this algorithm will escape this region by choosing a point with lower objective function value, which will be in a neighborhood of $x_0$. Algorithm 1 formally outlines this process.

---

**Algorithm 1** Deep Phase Retrieval (DPR) Gradient method

---

**Require:** Weights $W_i$, measurement matrix $A$, observations $|AG(x_0)|$, and step size $\alpha > 0$
1: Choose an arbitrary initial point $x_1 \in \mathbb{R}^k \setminus \{0\}$
2: **for** $i = 1, 2, \ldots$ **do**
3:     **if** $f(-x_i) < f(x_i)$ **then**
4:         $x_i \leftarrow -x_i$;
5:     **end if**
6:     Compute $v_{x_i, x_0} = (\Pi_{i=d}^1 W_{i,+,x_i})^\top A_{G(x_i)}^\top \left( |AG(x_i)| - |AG(x_0)| \right)$;
7:     $x_{i+1} = x_i - \alpha v_{x_i, x_0}$;
8: **end for**

---

**Remark.** We note that while the function is not differentiable, the descent direction is well-defined for all $x \in \mathbb{R}^k$ due to the definitions of $W_{i,+,x}$ and $A_{G(x)}$. When the objective function is differentiable, $v_{x, x_0}$ agrees with the true gradient. Otherwise, the descent direction only takes components of the formula for which the inputs to each ReLU are nonnegative.

## 3   Main Theoretical Analysis

We now formally present our main result. While the objective function is not smooth, its one-sided directional derivatives exist everywhere due to the continuity and piecewise linearity of $G$. Let $D_v f(x)$ denote the unnormalized one-sided directional derivative of $f$ at $x$ in the direction $v$: $D_v f(x) = \lim_{t \to 0^+} \frac{f(x+tv) - f(x)}{t}$.

**Theorem 2.** *Fix $\epsilon > 0$ such that $K_1 d^8 \epsilon^{1/4} \leqslant 1$ and let $d \geqslant 2$. Suppose $G$ is such that $W_i \in \mathbb{R}^{n_i \times n_{i-1}}$ has i.i.d. $\mathcal{N}(0, 1/n_i)$ entries for $i = 1, \ldots, d$. Suppose that $A \in \mathbb{R}^{m \times n_d}$ has i.i.d. $\mathcal{N}(0, 1/m)$*

*entries independent from $\{W_i\}$. Then if $m \geqslant C_\epsilon dk \log(n_1 n_2 \ldots n_d)$ and $n_i \geqslant C_\epsilon n_{i-1} \log n_{i-1}$ for $i = 1, \ldots, d$, then with probability at least $1 - \sum_{i=1}^d \gamma n_i e^{-c_\epsilon n_{i-1}} - \gamma m^{4k+1} e^{-c_\epsilon m}$, the following holds: for all non-zero $x, x_0 \in \mathbb{R}^k$, there exists $v_{x,x_0} \in \mathbb{R}^k$ such that the one-sided directional derivatives of $f$ satisfy*

$$D_{-v_{x,x_0}} f(x) < 0, \ \forall x \notin \mathcal{B}(x_0, K_2 d^3 \epsilon^{1/4} \|x_0\|) \cup \mathcal{B}(-\rho_d x_0, K_2 d^{14} \epsilon^{1/4} \|x_0\|) \cup \{0\},$$
$$D_x f(0) < 0, \ \forall x \neq 0,$$

*where $\rho_d > 0$ converges to 1 as $d \to \infty$ and $K_1$ and $K_2$ are universal constants. Here $C_\epsilon$ depends polynomially on $\epsilon^{-1}$, $c_\epsilon$ depends on $\epsilon$, and $\gamma$ is a universal constant.*

See Section 3.1 for the definition of the descent direction $v_{x,x_0}$. We note that while we assume the weights to have i.i.d. Gaussian entries, we make no assumption about the independence between layers. The result will be shown by proving the sufficiency of two deterministic conditions on the weights $W_i$ of our generative network and the measurement matrix $A$.

**Weight Distribution Condition.** The first condition quantifies the Gaussianity and spatial arrangement of the neurons in each layer. We say that $W \in \mathbb{R}^{n \times k}$ satisfies the *Weight Distribution Condition* (WDC) with constant $\epsilon > 0$ if for any non-zero $x, y \in \mathbb{R}^k$:

$$\left\| W_{+,x}^\top W_{+,y} - Q_{x,y} \right\| \leqslant \epsilon \text{ where } Q_{x,y} := \frac{\pi - \theta_{x,y}}{2\pi} I_k + \frac{\sin \theta_{x,y}}{2\pi} M_{\hat{x} \leftrightarrow \hat{y}}.$$

Here $\theta_{x,y} = \angle(x, y)$ and $M_{\hat{x} \leftrightarrow \hat{y}}{}^3$ is the matrix that sends $\hat{x} \mapsto \hat{y}$, $\hat{y} \mapsto \hat{x}$, and $z \mapsto 0$ for any $z \in \text{span}(\{x, y\})^\perp$. If $W_{ij} \sim \mathcal{N}(0, 1/n)$, then an elementary calculation gives $\mathbb{E}\left[ W_{+,x}^\top W_{+,y} \right] = Q_{x,y}$. [20] proved that Gaussian $W$ satisfies the WDC with high probability (Lemma 1 in the Appendix).

**Range Restricted Concentration Property.** The second condition is similar in the sense that it quantifies whether the measurement matrix behaves like a Gaussian when acting on the difference of pairs of vectors given by the output of the generative model. We say that $A \in \mathbb{R}^{m \times n}$ satisfies the *Range Restricted Concentration Property* (RRCP) with constant $\epsilon > 0$ if for all non-zero $x, y \in \mathbb{R}^k$, the matrices $A_{G(x)}$ and $A_{G(y)}$ satisfy the following for all $x_1, x_2, x_3, x_4 \in \mathbb{R}^k$:

$$\left| \langle (A_{G(x)}^\top A_{G(y)} - \Phi_{G(x),G(y)})(G(x_1) - G(x_2)), G(x_3) - G(x_4) \rangle \right|$$
$$\leqslant 31\epsilon \|G(x_1) - G(x_2)\| \|G(x_3) - G(x_4)\|$$

where

$$\Phi_{z,w} := \frac{\pi - 2\theta_{z,w}}{\pi} I_n + \frac{2 \sin \theta_{z,w}}{\pi} M_{\hat{z} \leftrightarrow \hat{w}}.$$

If $A_{ij} \sim \mathcal{N}(0, 1/m)$, then for any $z, w \in \mathbb{R}^n$, a similar calculation for Gaussian $W$ gives $\mathbb{E}\left[ A_z^\top A_w \right] = \Phi_{z,w}$. In our work, we establish that Gaussian $A$ satisfies the RRCP with high probability. Please see Section 6 in the Appendix for a complete proof.

We emphasize that these two conditions are deterministic, meaning that other distributions could be considered. We now state our main deterministic result.

**Theorem 3.** *Fix $\epsilon > 0$ such that $K_1 d^8 \epsilon^{1/4} \leqslant 1$ and let $d \geqslant 2$. Suppose that $G$ is such that $W_i \in \mathbb{R}^{n_i \times n_{i-1}}$ satisfies the WDC with constant $\epsilon$ for all $i = 1, \ldots, d$. Suppose $A \in \mathbb{R}^{m \times n_d}$ satisfies the RRCP with constant $\epsilon$. Then the same conclusion as Theorem 2 holds.*

## 3.1 Proof sketch for Theorem 2

Before we outline the proof of Theorem 2, we specify the descent direction $v_{x,x_0}$. For any $x \in \mathbb{R}^k$ where $f$ is differentiable, we have that

$$\nabla f(x) = (\Pi_{i=d}^1 W_{i,+,x})^\top A_{G(x)}^\top A_{G(x)} (\Pi_{i=d}^1 W_{i,+,x}) x - (\Pi_{i=d}^1 W_{i,+,x})^\top A_{G(x)}^\top A_{G(x_0)} (\Pi_{i=d}^1 W_{i,+,x_0}) x_0.$$

This is precisely the descent direction specified in Algorithm 1, expanded with our notation. When $f$ is not differentiable at $x$, choose a direction $w$ such that $f$ is differentiable at $x + \delta w$ for sufficiently small $\delta > 0$. Such a direction $w$ exists by the piecewise linearity of the generative model $G$. In fact, not only is the function piecewise linear, each of the pieces is the intersection of a finite number of half spaces. Thus, with probability 1 any randomly chosen direction $w$ moves strictly into one piece, allowing for differentiability at $x + \delta w$ for sufficiently small $\delta$. We note that any such $w$ can be chosen arbitrarily. Hence we define our descent direction $v_{x,x_0}$ as

$$v_{x,x_0} = \begin{cases} \nabla f(x) & f \text{ differentiable at } x \in \mathbb{R}^k \\ \lim_{\delta \to 0^+} \nabla f(x + \delta w) & \text{otherwise.} \end{cases}$$

The following is a sketch of the proof of Theorem 2:

- By the WDC and RRCP, we have that the descent direction $v_{x,x_0}$ concentrates uniformly for all non-zero $x, x_0 \in \mathbb{R}^k$ around a particular vector $\overline{v}_{x,x_0}$ defined by equation (5) in the Appendix.
- The WDC establishes that $\overline{v}_{x,x_0}$ concentrates uniformly for all non-zero $x, x_0 \in \mathbb{R}^k$ around a continuous vector $h_{x,x_0}$ defined by equation (7) in the Appendix.
- A direct analysis shows that $h_{x,x_0}$ is only small in norm for $x \approx x_0$ and $x \approx -\rho_d x_0$. See Section 5.3 for a complete proof. Since $v_{x,x_0} \approx \overline{v}_{x,x_0} \approx h_{x,x_0}$, $v_{x,x_0}$ is also only small in norm in neighborhoods around $x_0$ and $-\rho_d x_0$, establishing Theorem 3.
- Gaussian $W_i$ and $A$ satisfy the WDC and RRCP with high probability (Lemma 1 and Proposition 2 in the Appendix).

Theorem 2 is a combination of Lemma 1, Proposition 2, and Theorem 3. The full proofs of these results can be found in the Appendix.

**Remark.** In comparison to the results in [20], considerable technical advances were needed in our case, including establishing concentration of $A_{G(x)}$ over the range of $G$. The quantity $A_{G(x)}$ acts like a spatially dependent sensing matrix, requiring a condition similar to the Restricted Isometry Property that must hold simultaneously over a finite number of subspaces given by the range$(G)$.

## 4   Experiments

In this section, we investigate the use of enforcing generative priors in phase retrieval tasks. We compared our results with the sparse truncated amplitude flow algorithm (SPARTA) [35] and three popular general phase retrieval methods: Fienup [15], Gerchberg Saxton [16], and Wirtinger Flow [8]. A MATLAB implementation of the SPARTA algorithm was made publicly available by the authors at `https://gangwg.github.io/SPARTA/`. We implemented the last three algorithms using the MATLAB phase retrieval library PhasePack [10]. While these methods are not intended for sparse recovery, we include them to serve as baselines.

### 4.1   Experiments for Gaussian signals

We first consider synthetic experiments using Gaussian measurements on Gaussian signals. In particular, we considered a two layer network given by $G(x) = \text{relu}(W_2\text{relu}(W_1 x))$ where each $W_i$ has i.i.d. $\mathcal{N}(0,1)$ entries for $i = 1, 2$. We set $k = 10$, $n_1 = 500$, and $n_2 = 1000$. We let the entries of $A \in \mathbb{R}^{m \times n_2}$ and $x_0 \in \mathbb{R}^k$ be i.i.d. $\mathcal{N}(0,1)$. We ran Algorithm 1 for 25 random instances of $(A, W_1, W_2, x_0)$. A reconstruction $x^\star$ is considered successful if the relative error $\|G(x^\star) - G(x_0)\|/\|G(x_0)\| \leqslant 10^{-3}$. We also compared our results with SPARTA. In this setting, we chose a $k = 10$-sparse $y_0 \in \mathbb{R}^{n_2}$, where the nonzero coefficients are i.i.d. $\mathcal{N}(0,1)$. As before, we ran SPARTA with 25 random instances of $(A, y_0)$ and considered a reconstruction $y^\star$ successful if $\|y^\star - y_0\|/\|y_0\| \leqslant 10^{-3}$. We also experimented with sparsity levels $k = 3, 5$. Figure 2 displays the percentage of successful trials for different ratios $m/n$ where $n = n_2 = 1000$ and $m$ is the number of measurements.

### 4.2   Experiments for MNIST and CelebA

We next consider image recovery tasks, where we use two different generative models for the MNIST and CelebA datasets. In each task, the goal is to recover an image $y_0 \in \mathbb{R}^n$ given $|Ay_0|$ where

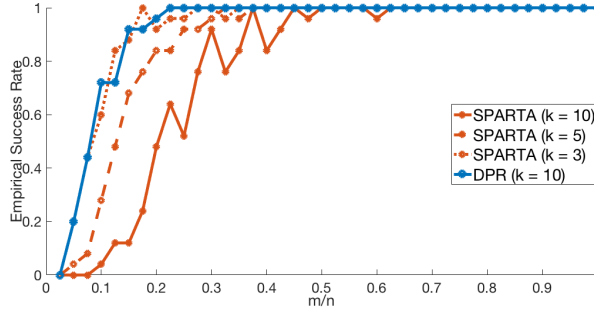

Figure 2: Empirical success rate with ratios $m/n$ where DPR's latent code dimension is $k = 10$, SPARTA's sparsity level ranges from $k = 3, 5$, and 10, and $n = 1000$. DPR achieves nearly the same empirical success rate of recovering a 10-dimensional latent code as SPARTA in recovering a 3-sparse 1000-dimensional signal.

$A \in \mathbb{R}^{m \times n}$ has i.i.d. $\mathcal{N}(0, 1/m)$ entries. We found an estimate image $G(x^\star)$ in the range of our generator via gradient descent, using the Adam optimizer [23]. Empirically, we noticed that Algorithm 1 would typically only negate the latent code (Lines 3–4) at the initial iterate, if necessary. Hence we use a modified version of Algorithm 1 in these image experiments: we ran two sessions of gradient descent for a random initial iterate $x_1$ and its negation $-x_1$ and chose the most successful reconstruction.

In the first image experiment, we used a pretrained Variational Autoencoder (VAE) from [4] that was trained on the MNIST dataset [24]. This dataset consists of $60,000$ images of handwritten digits. Each image is of size $28 \times 28$, resulting in vectorized images of size $784$. As described in [4], the recognition network is of size $784 - 500 - 500 - 20$ while the generator network is of size $20 - 500 - 500 - 784$. The latent code space dimension is $k = 20$.

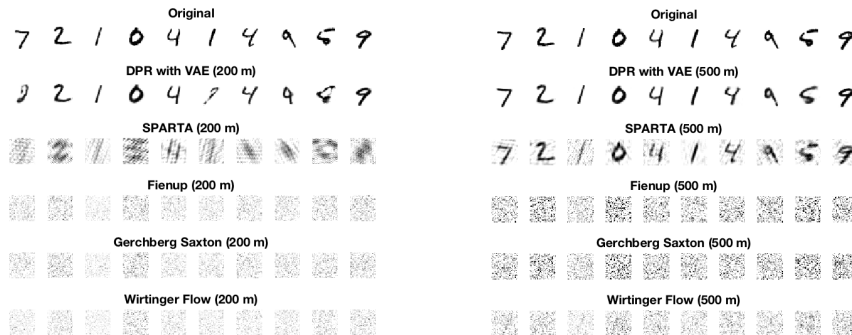

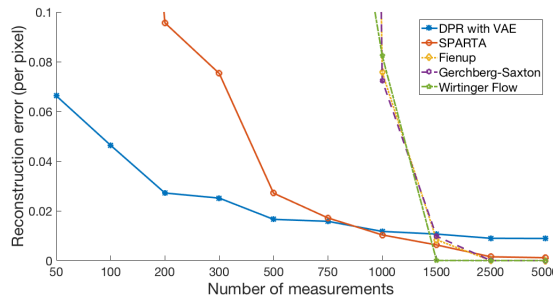

Figure 3: Top left: Example reconstructions with 200 measurements. Top right: Example reconstructions with 500 measurements. Bottom: A comparison of DPR's reconstruction error versus each algorithm for different numbers of measurements.

For SPARTA, we performed sparse recovery by transforming the images using the 2-D Discrete Cosine Transform (DCT). We allowed 10 random restarts for each algorithm, including the sparse and general phase retrieval methods. The results in Figure 3 demonstrate the success of our algorithm with very few measurements. For 200 measurements, we can achieve reasonable recovery. SPARTA can achieve good recovery with 500 measurements while the other algorithms cannot. In addition, our algorithm exhibits recovery with 500 measurements compared to the alternatives requiring 1000 and 1500 measurements, which is where they begin to succeed. The performance for the general phase retrieval methods is to be expected as they are known to succeed only when $m = \Omega(n)$ where $n = 784$.

We note that while our algorithm succeeds with fewer measurements than the other methods, our performance, as measured by per-pixel reconstruction error, saturates as the number of measurements increases since our reconstruction accuracy is ultimately bounded by the generative model's representational error. As generative models improve, their representational errors will decrease. Nonetheless, as can be seen in the reconstructed digits, the recoveries are semantically correct (the correct digit is legibly recovered) even though the reconstruction error does not decay to zero. In applications, such as MRI and molecular structure estimation via X-ray crystallography, semantic error measures would be more informative estimates of recovery performance than per-pixel error measures.

In the second experiment, we used a pretrained Deep Convolutional Generative Adversarial Network (DCGAN) from [4] that was trained on the CelebA dataset [27]. This dataset consists of $200,000$ facial images of celebrities. The RGB images were cropped to be of size $64 \times 64$, resulting in vectorized images of dimension $64 \times 64 \times 3 = 12288$. The latent code space dimension is $k = 100$. We allowed 2 random restarts. We ran numerical experiments with the other methods and they did not succeed at measurement levels below 5000. The general phase retrieval methods began reconstructing the images when $m = \Omega(n)$ where $n = 12288$. The following figure showcases our results on reconstructing 10 images from the DCGAN's test set with 500 measurements.

Original

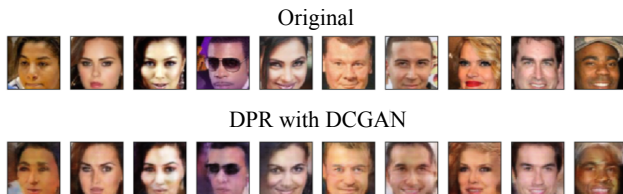

DPR with DCGAN

Figure 4: 10 reconstructed images from celebA's test set using DPR with 500 measurements.

## Acknowledgments

OL acknowledges support by the NSF Graduate Research Fellowship under Grant No. DGE-1450681. PH acknowledges funding by the grant NSF DMS-1464525.

## Footnotes

[2] with respect to the dimensionality of the latent code given to the generative network

[3] A formula for this matrix is as follows: consider a rotation matrix $R$ that sends $\hat{x} \mapsto e_1$ and $\hat{y} \mapsto \cos \theta_0 e_1 + \sin \theta_0 e_2$ where $\theta_0 = \angle(x, y)$. Then $M_{\hat{x} \leftrightarrow \hat{y}} = R^\top \begin{bmatrix} \cos \theta_0 & \sin \theta_0 & 0 \\ \sin \theta_0 & -\cos \theta_0 & 0 \\ 0 & 0 & 0_{k-2} \end{bmatrix} R$ where $0_{k-2}$ is the $k-2 \times k-2$ matrix of zeros. Note that if $\theta_0 = 0$ or $\pi$, $M_{\hat{x} \leftrightarrow \hat{y}} = \hat{x}\hat{x}^\top$ or $-\hat{x}\hat{x}^\top$, respectively.

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
