[Supplementary Material · DPR NeurIPS 2018 Final Supplementary.pdf]

# 5 Appendix A: Results for Main Theorem

**Notation.** Let $(\cdot)^\top$ denote the real transpose. Let $[n] = \{1, \ldots, n\}$. Let $\mathcal{B}(x, r)$ denote the Euclidean ball centered at $x$ with radius $r$. Let $\|\cdot\|$ denote the $\ell_2$ norm for vectors and spectral norm for matrices. For any non-zero $x \in \mathbb{R}^n$, let $\hat{x} = x/\|x\|$. Let $\Pi_{i=d}^1 W_i = W_d W_{d-1} \ldots W_1$. Let $I_n$ be the $n \times n$ identity matrix. Let $\mathcal{S}^{k-1}$ denote the unit sphere in $\mathbb{R}^k$. We write $c = \Omega(\delta)$ when $c \geqslant C\delta$ for some positive constant $C$. Similarly, we write $c = O(\delta)$ when $c \leqslant C\delta$ for some positive constant $C$. When we say that a constant depends polynomially on $\epsilon^{-1}$, this means that it is at least $C\epsilon^{-k}$ for some positive $C$ and positive integer $k$. For notational convenience, we write $a = b + O_1(\epsilon)$ if $\|a - b\| \leqslant \epsilon$ where $\|\cdot\|$ denotes $|\cdot|$ for scalars, $\ell_2$ norm for vectors, and spectral norm for matrices. Define $\mathrm{sgn} : \mathbb{R} \to \mathbb{R}$ to be $\mathrm{sgn}(x) = x/|x|$ for non-zero $x \in \mathbb{R}$ and $\mathrm{sgn}(x) = 0$ otherwise. For a vector $v \in \mathbb{R}^n$, $\mathrm{diag}(\mathrm{sgn}(v))$ is $\mathrm{sgn}(v_i)$ in the $i$-th diagonal entry and $\mathrm{diag}(v > 0)$ is 1 in the $i$-th diagonal entry if $v_i > 0$ and 0 otherwise. For non-zero $x, x_0 \in \mathbb{R}^k$, let $\theta_0 = \angle(x, x_0)$. To understand how the map $x \mapsto \mathrm{relu}(Wx)$ distorts angles in expectation, define $g : \mathbb{R} \to \mathbb{R}$ by

$$g(\theta) = \cos^{-1}\left(\frac{\cos\theta(\pi - \theta) + \sin\theta}{\pi}\right).$$

Then for $i \geqslant 1$, set $\overline{\theta}_i = g(\overline{\theta}_{i-1})$ where $\overline{\theta}_0 = \theta_0$. Let $g^{\circ d}$ denote the composition of $g$ with itself $d$ times. In this section, $L$ is the positive universal constant $3 + 88/\pi$.[4]

## 5.1 Full proof of Theorem 3

*Proof.* Set

$$v_{x,x_0} = \begin{cases} \nabla f(x) & f \text{ is differentiable at } x \in \mathbb{R}^k \\ \lim_{\delta \to 0^+} \nabla f(x + \delta w) & \text{otherwise,} \end{cases}$$

where $f$ is differentiable at $x + \delta w$ for sufficiently small $\delta > 0$. Any such direction $w$ can be chosen arbitrarily. Recall that

$$\nabla f(x) = (\Pi_{i=d}^1 W_{i,+,x})^\top A_{G(x)}^\top A_{G(x)}(\Pi_{i=d}^1 W_{i,+,x})x - (\Pi_{i=d}^1 W_{i,+,x})^\top A_{G(x)}^\top A_{G(x_0)}(\Pi_{i=d}^1 W_{i,+,x_0})x_0.$$

Let

$$\overline{v}_{x,x_0} := (\Pi_{i=d}^1 W_{i,+,x})^\top (\Pi_{i=d}^1 W_{i,+,x})x - (\Pi_{i=d}^1 W_{i,+,x})^\top \Phi_{G(x),G(x_0)}(\Pi_{i=d}^1 W_{i,+,x_0})x_0, \quad (5)$$

$$h_{x,x_0} := -\frac{\|x_0\|}{2^d}\left(\frac{\pi - 2\overline{\theta}_d}{\pi}\right)\left(\prod_{i=0}^{d-1}\frac{\pi - \overline{\theta}_i}{\pi}\right)\hat{x}_0 \tag{6}$$

$$+ \frac{1}{2^d}\left[\|x\| - \|x_0\|\left(\frac{2\sin\overline{\theta}_d}{\pi} + \left(\frac{\pi - 2\overline{\theta}_d}{\pi}\right)\sum_{i=0}^{d-1}\frac{\sin\overline{\theta}_i}{\pi}\left(\prod_{j=i+1}^{d-1}\frac{\pi - \overline{\theta}_j}{\pi}\right)\right)\right]\hat{x}, \tag{7}$$

and

$$S_{\epsilon,x_0} := \left\{x \in \mathbb{R}^k \setminus \{0\} : \|h_{x,x_0}\| \leqslant \frac{1}{2^d}\epsilon\max(\|x\|, \|x_0\|)\right\}.$$

First, observe that by the WDC, we have that for all $x \neq 0$ and $i = 1, \ldots, d$,

$$\left\|W_{i,+,x}^\top W_{i,+,x} - \frac{1}{2}I_{n_i}\right\| \leqslant \epsilon \implies \|W_{i,+,x}\|^2 \leqslant \frac{1}{2} + \epsilon. \tag{8}$$

Observe that

$$\|\nabla f(x) - \overline{v}_{x,x_0}\| \leqslant \left\|(\Pi_{i=d}^1 W_{i,+,x})^\top (A_{G(x)}^\top A_{G(x)} - I_{n_d})(\Pi_{i=d}^1 W_{i,+,x})x\right\|$$

$$+ \left\|(\Pi_{i=d}^1 W_{i,+,x})^\top (A_{G(x)}^\top A_{G(x_0)} - \Phi_{G(x),G(x_0)})(\Pi_{i=d}^1 W_{i,+,x_0})x_0\right\|.$$

Hence by the RRCP (Proposition 6) and (8), we have that

$$\|\nabla f(x) - \overline{v}_{x,x_0}\| \leqslant L\epsilon \left( \prod_{i=1}^{d} \|W_{i,+,x}\|^2 + \prod_{i=1}^{d} \|W_{i,+,x}\| \|W_{i,+,x_0}\| \right) \max(\|x\|, \|x_0\|) \quad (9)$$

$$\leqslant 2L\epsilon \left( \frac{1}{2} + \epsilon \right)^d \max(\|x\|, \|x_0\|). \quad (10)$$

Then Lemma 2 guarantees that for all non-zero $x, x_0 \in \mathbb{R}^k$,

$$\|\overline{v}_{x,x_0} - h_{x,x_0}\| \leqslant 78 \frac{d^3}{2^d} \sqrt{\epsilon} \max(\|x\|, \|x_0\|). \quad (11)$$

Then we have that for all non-zero $x, x_0 \in \mathbb{R}^k$,

$$\|v_{x,x_0} - h_{x,x_0}\| = \lim_{\delta \to 0^+} \|\nabla f(x + \delta w) - h_{x+\delta w, x_0}\|$$

$$\leqslant \lim_{\delta \to 0^+} \left( \|\nabla f(x + \delta w) - \overline{v}_{x+\delta w, x_0}\| + \|\overline{v}_{x+\delta w, x_0} - h_{x+\delta w, x_0}\| \right)$$

$$\leqslant \sqrt{\epsilon} \left( 2L \frac{(1 + 2\epsilon)^d}{2^d} + 78 \frac{d^3}{2^d} \right) \max(\|x\|, \|x_0\|)$$

$$\leqslant \sqrt{\epsilon} K \frac{d^3}{2^d} \max(\|x\|, \|x_0\|)$$

for some universal constant $K$ where the first equality follows by the definition of $v_{x,x_0}$ and the continuity of $h_{x,x_0}$ for non-zero $x, x_0$. The second inequality combines (10) and (11) and since $2\epsilon d \leqslant 1 \implies (1 + 2\epsilon)^d \leqslant e^{2\epsilon d} \leqslant 1 + 4\epsilon d$. This establishes concentration of $v_{x,x_0}$ to $h_{x,x_0}$ for all non-zero $x, x_0 \in \mathbb{R}^k$:

$$\|v_{x,x_0} - h_{x,x_0}\| \leqslant \sqrt{\epsilon} K \frac{d^3}{2^d} \max(\|x\|, \|x_0\|) \quad (12)$$

Now, due to the continuity and piecewise linearity of the function $G(x)$ and $|\cdot|$, we have that for any $x, y \neq 0$ that there exists a sequence $\{x_n\} \to x$ such that $f$ is differentiable at each $x_n$ and $D_y f(x) = \lim_{n \to \infty} \nabla f(x_n) \cdot y$. Thus, as $\nabla f(x_n) = v_{x_n, x_0}$,

$$D_{-v_{x,x_0}} f(x) = - \lim_{n \to \infty} v_{x_n, x_0} \cdot v_{x,x_0}.$$

Then observe that

$$v_{x_n,x_0} \cdot v_{x,x_0} = h_{x_n,x_0} \cdot h_{x,x_0} + (v_{x_n,x_0} - h_{x_n,x_0}) \cdot h_{x,x_0} + h_{x_n,x_0} \cdot (v_{x,x_0} - h_{x,x_0})$$

$$+ (v_{x_n,x_0} - h_{x_n,x_0}) \cdot (v_{x,x_0} - h_{x,x_0})$$

$$\geqslant h_{x_n,x_0} \cdot h_{x,x_0} - \|v_{x_n,x_0} - h_{x_n,x_0}\| \|h_{x,x_0}\| - \|h_{x_n,x_0}\| \|v_{x,x_0} - h_{x,x_0}\|$$

$$- \|v_{x_n,x_0} - h_{x_n,x_0}\| \|v_{x,x_0} - h_{x,x_0}\|$$

$$\geqslant h_{x_n,x_0} \cdot h_{x,x_0} - \|h_{x,x_0}\| \sqrt{\epsilon} K \frac{d^3}{2^d} \max(\|x\|, \|x_0\|)$$

$$- \|h_{x_n,x_0}\| \sqrt{\epsilon} K \frac{d^3}{2^d} \max(\|x\|, \|x_0\|) - \epsilon \left[ K \frac{d^3}{2^d} \right]^2 \max(\|x_n\|, \|x_0\|) \max(\|x\|, \|x_0\|)$$

where in the last inequality, we used (12). By the continuity of $h_{x,x_0}$ for non-zero $x \in \mathbb{R}^k$, we have that for $x \in S^c_{4\sqrt{\epsilon} K d^3, x_0}$:

$$\lim_{n \to \infty} v_{x_n,x_0} \cdot v_{x,x_0} \geqslant \|h_{x,x_0}\|^2 - 2\|h_{x,x_0}\| \sqrt{\epsilon} K \frac{d^3}{2^d} \max(\|x\|, \|x_0\|) - \epsilon \left[ K \frac{d^3}{2^d} \right]^2 \max(\|x\|, \|x_0\|)^2$$

$$= \frac{\|h_{x,x_0}\|}{2} \left( \|h_{x,x_0}\| - 4\sqrt{\epsilon} K \frac{d^3}{2^d} \max(\|x\|, \|x_0\|) \right)$$

$$+ \frac{1}{2} \left( \|h_{x,x_0}\|^2 - 2\epsilon \left[ K \frac{d^3}{2^d} \right]^2 \max(\|x\|, \|x_0\|)^2 \right)$$

$$> 0.$$

Hence we conclude that for all $x \in S^c_{4\sqrt{\epsilon K d^3}, x_0}$, $D_{-v_{x,x_0}} f(x) < 0$.

We now show that $D_x f(0) < 0$ for all $x \neq 0$. Observe that we can write the objective function as

$$f(x) = \frac{1}{2} \sum_{\ell=1}^m \left( |\langle a_\ell, (\Pi^1_{i=d} W_{i,+,x}) x \rangle| - |\langle a_\ell, (\Pi^1_{i=d} W_{i,+,x_0}) x_0 \rangle| \right)^2$$

where $a_\ell$ is a row of $A$. Then for any $t > 0$, we have that by the positive homogeneity of $G$,

$$f(tx) = \frac{1}{2} \sum_{\ell=1}^m (t^2 |\langle a_\ell, (\Pi^1_{i=d} W_{i,+,x}) x \rangle|^2 + |\langle a_\ell, (\Pi^1_{i=d} W_{i,+,x_0}) x_0 \rangle|^2$$
$$- 2t |\langle a_\ell, (\Pi^1_{i=d} W_{i,+,x}) x \rangle \langle a_\ell, (\Pi^1_{i=d} W_{i,+,x_0}) x_0 \rangle|).$$

Then since

$$f(0) = \frac{1}{2} \sum_{\ell=1}^m |\langle a_\ell, (\Pi^1_{i=d} W_{i,+,x_0}) x_0 \rangle|^2$$

we have that

$$D_x f(0) = \lim_{t \to 0^+} \frac{f(tx) - f(0)}{t}$$
$$= -\sum_{\ell=1}^m |\langle a_\ell, (\Pi^1_{i=d} W_{i,+,x}) x \rangle \langle a_\ell, (\Pi^1_{i=d} W_{i,+,x_0}) x_0 \rangle|$$
$$= -\langle (\Pi^1_{i=d} W_{i,+,x}) x, A^\top_{G(x)} A_{G(x_0)} (\Pi^1_{i=d} W_{i,+,x_0}) x_0 \rangle.$$

We now focus on bounding this quantity from above by using the angle concentration property derived in Lemma 4. We use the shorthand notation $\Lambda_x := \Pi^1_{i=d} W_{i,+,x}$ and $\Lambda_{x_0} := \Pi^1_{i=d} W_{i,+,x_0}$. Observe that we can write

$$\langle \Lambda_x x, A^\top_{G(x)} A_{G(x_0)} \Lambda_{x_0} x_0 \rangle = \cos(\angle (A_{G(x)} \Lambda_x x, A_{G(x_0)} \Lambda_{x_0} x_0)) \| A_{G(x_0)} \Lambda_x x \| \| A_{G(x_0)} \Lambda_{x_0} x_0 \|. \tag{13}$$

However, by Lemma 4, we have that

$$\cos \varphi(\theta_d) - 4L\epsilon \leqslant \cos(\angle (A_{G(x)} \Lambda_x x, A_{G(x_0)} \Lambda_{x_0} x_0)) \leqslant \cos \varphi(\theta_d) + 4L\epsilon \tag{14}$$

where $\varphi$ is defined in (24) and $\theta_d := \angle (\Lambda_x x, \Lambda_{x_0} x_0)$. Thus combining (14) and (13) gives

$$\langle \Lambda_x x, A^\top_{G(x)} A_{G(x_0)} \Lambda_{x_0} x_0 \rangle \geqslant (\cos \varphi(\theta_d) - 4L\epsilon) \| A_{G(x)} \Lambda_x x \| \| A_{G(x_0)} \Lambda_{x_0} x_0 \|. \tag{15}$$

However, note that

$$\cos \varphi(\theta) = \frac{(\pi - 2\theta) \cos \theta + 2 \sin \theta}{\pi} \geqslant \frac{2}{\pi} \ \forall \, \theta \in [0, \pi]. \tag{16}$$

Hence if $\epsilon < 1/(4L\pi)$, we have that by (15), (16), and (13), the following holds:

$$\langle \Lambda_x x, A^\top_{G(x)} A_{G(x_0)} \Lambda_{x_0} x_0 \rangle \geqslant \frac{1}{\pi} \| A_{G(x)} \Lambda_x x \| \| A_{G(x_0)} \Lambda_{x_0} x_0 \|. \tag{17}$$

Finally, Lemma 4 establishes that for all non-zero $x, x_0 \in \mathbb{R}^k$,

$$\| A_{G(x)} \Lambda_x x \|, \| A_{G(x_0)} \Lambda_{x_0} x_0 \| \neq 0. \tag{18}$$

Hence we conclude that

$$D_x f(0) = -\langle \Lambda_x x, A^\top_{G(x)} A_{G(x_0)} \Lambda_{x_0} x_0 \rangle$$
$$\leqslant -\frac{1}{\pi} \| A_{G(x)} \Lambda_x x \| \| A_{G(x_0)} \Lambda_{x_0} x_0 \|$$
$$< 0$$

where we used (17) in the first inequality and (18) in the last inequality.

We conclude by applying Proposition 1 and $24\pi d^6 \sqrt{4\sqrt{\epsilon K d^3}} \leqslant 1$ to attain

$$S_{4\sqrt{\epsilon K d^3}, x_0} \subset \mathcal{B}(x_0, 89d\sqrt{4\sqrt{\epsilon K d^3}} \| x_0 \|) \cup \mathcal{B}(\rho_d x_0, 77422\pi^2 d^{12} \sqrt{4\sqrt{\epsilon K d^3}} \| x_0 \|).$$

$\square$

We record some results that were used in the above proof. In [20], it was shown that Gaussian $W_i$ satisfies the WDC with high probability:

**Lemma 1** (Lemma 9 in [20]). *Fix $0 < \epsilon < 1$. Let $W \in \mathbb{R}^{n \times k}$ have i.i.d. $\mathcal{N}(0, 1/n)$ entries. If $n \geqslant ck \log k$ then with probability at least $1 - 8n \exp(-\gamma k)$, $W$ satisfies the WDC with constant $\epsilon$. Here $c, \gamma^{-1}$ are constants that depend only polynomially on $\epsilon^{-1}$.*

The following is a technical result showing concentration of $\overline{v}_{x,x_0}$ around $h_{x,x_0}$:

**Lemma 2.** *Fix $0 < \epsilon < d^{-4}(1/16\pi)^2$ and let $d \geqslant 2$. Let $W_i$ satisfy the WDC with constant $\epsilon$ for $i = 1, \ldots d$. For any non-zero $x, y \in \mathbb{R}^k$, we have*

$$\|\overline{v}_{x,y} - h_{x,y}\| \leqslant \frac{78d^3}{2^d} \sqrt{\epsilon} \max(\|x\|, \|y\|).$$

*Proof.* Observe that

$$\|\overline{v}_{x,y} - h_{x,y}\| \leqslant \underbrace{\left\| (\Pi_{i=d}^1 W_{i,+,x})^\top (\Pi_{i=d}^1 W_{i,+,x}) x - \frac{1}{2^d} x \right\|}_{=Q_1}$$

$$+ \underbrace{\left\| \frac{\pi - 2\theta_d}{\pi} (\Pi_{i=d}^1 W_{i,+,x})^\top (\Pi_{i=d}^1 W_{i,+,y}) y - \frac{\pi - 2\overline{\theta}_d}{\pi} \tilde{h}_{x,y} \right\|}_{=Q_2}$$

$$+ \underbrace{\left\| \frac{2 \sin \theta_d}{\pi} \frac{\|(\Pi_{i=d}^1 W_{i,+,y}) y\|}{\|(\Pi_{i=d}^1 W_{i,+,x}) x\|} (\Pi_{i=d}^1 W_{i,+,x})^\top (\Pi_{i=d}^1 W_{i,+,x}) x - \frac{2 \sin \overline{\theta}_d}{\pi} \frac{\|y\|}{\|x\|} \frac{1}{2^d} x \right\|}_{=Q_3}.$$

We focus on bounding each individual quantity $Q_i$ for $i = 1, 2, 3$. For $Q_1$, we have that by (20) in Lemma 3,

$$Q_1 = \left\| (\Pi_{i=d}^1 W_{i,+,x})^\top (\Pi_{i=d}^1 W_{i,+,x}) x - \frac{1}{2^d} x \right\| \leqslant 24 \frac{d^3 \sqrt{\epsilon}}{2^d} \|x\|. \tag{19}$$

Then for $Q_2$, observe that by the triangle inequality, we have

$$Q_2 \leqslant \left\| \frac{\pi - 2\theta_d}{\pi} (\Pi_{i=d}^1 W_{i,+,x})^\top (\Pi_{i=d}^1 W_{i,+,y}) y - \frac{\pi - 2\theta_d}{\pi} \tilde{h}_{x,y} \right\|$$

$$+ \left\| \frac{\pi - 2\theta_d}{\pi} \tilde{h}_{x,y} - \frac{\pi - 2\overline{\theta}_d}{\pi} \tilde{h}_{x,y} \right\|$$

$$\overset{(*)}{\leqslant} \left| \frac{\pi - 2\theta_d}{\pi} \right| 24 \frac{d^3 \sqrt{\epsilon}}{2^d} \|y\| + \left| \frac{2}{\pi} (\theta_d - \overline{\theta}_d) \right| \|\tilde{h}_{x,y}\|$$

$$\overset{(**)}{\leqslant} 24 \frac{d^3 \sqrt{\epsilon}}{2^d} \|y\| + \frac{8d\sqrt{\epsilon}}{\pi} \frac{(1 + \frac{d}{\pi})}{2^d} \|y\|$$

where in $(*)$ we used (20) and $(**)$ used (23) and the fact that $\|\tilde{h}_{x,y}\| \leqslant 2^{-d}(1 + \frac{d}{\pi}) \|y\|$. Hence

$$Q_2 \leqslant \frac{1}{2^d} \left( 24d^3 + \frac{8d}{\pi} \left( 1 + \frac{d}{\pi} \right) \right) \sqrt{\epsilon} \|y\|.$$

To bound $Q_3$, let $y_d := (\Pi_{i=d}^1 W_{i,+,y})y$ and $x_d := (\Pi_{i=d}^1 W_{i,+,x})x$. We use the triangle inequality to gather the following three quantities to bound:

$$Q_3 \leqslant \underbrace{\left| \frac{2\sin\theta_d}{\pi} - \frac{2\sin\overline{\theta}_d}{\pi} \right| \frac{\|y_d\|}{\|x_d\|} \left\| (\Pi_{i=d}^1 W_{i,+,x})^\top x_d \right\|}_{=Q_{3,1}}$$

$$+ \underbrace{\left\| \frac{2\sin\overline{\theta}_d}{\pi} \frac{\|y_d\|}{\|x_d\|} (\Pi_{i=d}^1 W_{i,+,x})^\top (\Pi_{i=d}^1 W_{i,+,x})x - \frac{2\sin\overline{\theta}_d}{\pi} \frac{\|y\|}{\|x\|} (\Pi_{i=d}^1 W_{i,+,x})^\top (\Pi_{i=d}^1 W_{i,+,x})x \right\|}_{=Q_{3,2}}$$

$$+ \underbrace{\left\| \frac{2\sin\overline{\theta}_d}{\pi} \frac{\|y\|}{\|x\|} (\Pi_{i=d}^1 W_{i,+,x})^\top (\Pi_{i=d}^1 W_{i,+,x})x - \frac{2\sin\overline{\theta}_d}{\pi} \frac{\|y\|}{\|x\|} \frac{1}{2^d}x \right\|}_{=Q_{3,3}}.$$

Using (8) and (23) gives

$$Q_{3,1} \leqslant \frac{2}{\pi} |\theta_d - \overline{\theta}_d| \left( \frac{1}{2} + \epsilon \right)^d \frac{\|y\|}{\|x\|} \|x\|$$

$$\leqslant \frac{8d}{\pi} \left( \frac{1}{2} + \epsilon \right)^d \sqrt{\epsilon}\|y\|$$

$$= \frac{8d(1+2\epsilon)^d}{\pi 2^d} \sqrt{\epsilon}\|y\|.$$

Likewise, equations (8) and (22) gives

$$Q_{3,2} \leqslant \left| \frac{\|y_d\|}{\|x_d\|} - \frac{\|y\|}{\|x\|} \right| \left| \frac{2\sin\overline{\theta}_d}{\pi} \right| \left( \frac{1}{2} + \epsilon \right)^d \|x\|$$

$$\leqslant 8d\epsilon \frac{\|y\|}{\|x\|} \frac{2}{\pi} \left( \frac{1}{2} + \epsilon \right)^d \|x\|$$

$$\leqslant \frac{16d\sqrt{\epsilon}}{\pi} \left( \frac{1}{2} + \epsilon \right)^d \|y\|$$

$$= \frac{16d(1+2\epsilon)^d}{\pi 2^d} \sqrt{\epsilon}\|y\|$$

Lastly, we use (20) to attain

$$Q_{3,3} \leqslant \left| \frac{2\sin\overline{\theta}_d}{\pi} \right| \frac{\|y\|}{\|x\|} \left\| (\Pi_{i=d}^1 W_{i,+,x})^\top (\Pi_{i=d}^1 W_{i,+,x})x - \frac{1}{2^d}x \right\|$$

$$\leqslant \frac{2}{\pi} \frac{\|y\|}{\|x\|} 24 \frac{d^3\sqrt{\epsilon}}{2^d} \|x\|$$

$$\leqslant \frac{48d^3\sqrt{\epsilon}}{\pi 2^d} \|y\|.$$

Combining the bounds for $Q_{3,i}$ for $i = 1, 2, 3$ gives

$$Q_3 \leqslant Q_{3,1} + Q_{3,2} + Q_{3,3}$$

$$\leqslant \frac{8d(1+2\epsilon)^d}{\pi 2^d} \sqrt{\epsilon}\|y\| + \frac{16d(1+2\epsilon)^d}{\pi 2^d} \sqrt{\epsilon}\|y\| + \frac{48d^3\sqrt{\epsilon}}{\pi 2^d} \|y\|$$

$$= \frac{1}{2^d} \left( \frac{24d(1+2\epsilon)^d + 48d^3}{\pi} \right) \sqrt{\epsilon}\|y\|.$$

Thus we attain

$$Q_1 + Q_2 + Q_3 \leqslant \frac{K_d}{2^d} \sqrt{\epsilon} \max(\|x\|, \|y\|)$$

where

$$K_d = 24d^3 + 24d^3 + \frac{8d(1+d/\pi)}{\pi} + \frac{24(1+2\epsilon)^d}{\pi} + \frac{48d^3}{\pi} \leqslant 78d^3$$

as long as $\epsilon \leqslant \min(1/2d, 1/96)$.  □

The following result summarizes some useful bounds from [20]:

**Lemma 3** (Results from Lemma 5 in [20]). *Fix $0 < \epsilon < d^{-4}(1/16\pi)^2$ and let $d \geqslant 2$. Let $W_i$ satisfy the WDC with constant $\epsilon$ for $i = 1, \ldots d$. Then for any non-zero $x, y \in \mathbb{R}^k$, the following hold:*

$$\left\| (\Pi_{i=d}^1 W_{i,+x})^\top (\Pi_{i=d}^1 W_{i,+,y}) y - \tilde{h}_{x,y} \right\| \leqslant 24 \frac{d^3 \sqrt{\epsilon}}{2^d} \|y\|, \tag{20}$$

$$\left\langle (\Pi_{i=d}^1 W_{i,+,x}) x, (\Pi_{i=d}^1 W_{i,+,y}) y \right\rangle \geqslant \frac{1}{4\pi} \frac{1}{2^d} \|x\| \|y\|, \tag{21}$$

$$\left| \frac{\|y_d\|}{\|x_d\|} - \frac{\|y\|}{\|x\|} \right| \leqslant 8d\epsilon \frac{\|y\|}{\|x\|}, \tag{22}$$

$$|\theta_d - \overline{\theta}_d| \leqslant 4d\sqrt{\epsilon} \tag{23}$$

*where $x_d := (\Pi_{i=d}^1 W_{i,+x}) x$, $y_d := (\Pi_{i=d}^1 W_{i,+y}) y$, $\theta_d := \angle(x_d, y_d)$, $\overline{\theta}_d := g^{\circ d}(\angle(x, y))$, and the vector $\tilde{h}_{x,y}$ is defined as*

$$\tilde{h}_{x,y} := \frac{1}{2^d} \left[ \left( \prod_{i=0}^{d-1} \frac{\pi - \overline{\theta}_i}{\pi} \right) y + \sum_{i=0}^{d-1} \frac{\sin \overline{\theta}_i}{\pi} \left( \prod_{j=i+1}^{d-1} \frac{\pi - \overline{\theta}_j}{\pi} \right) \frac{\|y\|}{\|x\|} x \right].$$

.

## 5.2 Angle Concentration Property of $A_{G(x)}$

We need to understand how the operator $z \mapsto A_z z$ distorts angles. Observe that for $z, w \in \mathcal{S}^{n-1}$ for which the RRCP holds, we have that

$$
\begin{aligned}
\langle z, A_z^\top A_w w \rangle \approx \langle z, \Phi_{z,w} w \rangle &= \left\langle z, \left( \frac{\pi - 2\theta_{z,w}}{\pi} I + \frac{2 \sin \theta_{z,w}}{\pi} M_{z \leftrightarrow w} \right) w \right\rangle \\
&= \frac{\pi - 2\theta_{z,w}}{\pi} \langle z, w \rangle + \frac{2 \sin \theta_{z,w}}{\pi} \|z\|^2 \\
&= \frac{(\pi - 2\theta_{z,w}) \cos \theta_{z,w} + 2 \sin \theta_{z,w}}{\pi} \\
&:= \cos \varphi(\theta_{z,w})
\end{aligned}
$$

where $\varphi : \mathbb{R} \to \mathbb{R}$ is defined by

$$\varphi(\theta) := \cos^{-1} \left( \frac{(\pi - 2\theta) \cos \theta + 2 \sin \theta}{\pi} \right). \tag{24}$$

The following lemma establishes that the angle $\angle(A_{G(x)} G(x), A_{G(y)} G(y))$ concentrates around $\varphi(\angle(G(x), G(y)))$.

**Lemma 4.** *Fix $0 < \epsilon < 1/4L$. Suppose $A \in \mathbb{R}^{m \times n_d}$ satisfies the RRCP with constant $\epsilon$. Suppose $G$ is such that each $W_i \in \mathbb{R}^{n_i \times n_{i-1}}$ satisfy the WDC with constant $\epsilon$ for all $i \in [d]$. Then for all $x, y \in \mathbb{R}^k \setminus \{0\}$, the angle $\theta_1 := \angle(A_{G(x)} G(x), A_{G(y)} G(y))$ is well-defined and*

$$|\cos \theta_1 - \cos \varphi(\theta_0)| \leqslant 4L\epsilon$$

*where $\theta_0 = \angle(G(x), G(y))$, $\varphi$ is defined in (24), and $L$ is a positive universal constant.*

*Proof.* Fix $x, y \in \mathbb{R}^k \setminus \{0\}$. We use the shorthand notation $\Lambda_x := \Pi_{i=d}^1 W_{i,+,x}$ and $\Lambda_y := \Pi_{i=d}^1 W_{i,+,y}$. Note that the WDC implies that for sufficiently small $\epsilon$, we have that $\Lambda_x x, \Lambda_y y \neq 0$.

Hence we may assume, without loss of generality, that $\|\Lambda_x x\| = \|\Lambda_y y\| = 1$. Now define the following quantities:

$$\delta_1 := \langle \Lambda_x x, (A_{G(x)}^\top A_{G(y)} - \Phi_{G(x),G(y)})\Lambda_y y \rangle,$$
$$\delta_2 := \langle \Lambda_x x, (A_{G(x)}^\top A_{G(x)} - I)\Lambda_x x \rangle$$
$$\delta_3 := \langle \Lambda_y y, (A_{G(y)}^\top A_{G(y)} - I)\Lambda_y y \rangle.$$

Observe that by the RRCP, we have that $\max_{i=1,2,3} |\delta_i| \leqslant L\epsilon$. Hence if $0 < \epsilon < 1/L$,

$$0 < 1 - L\epsilon \leqslant \|A_{G(x)}\Lambda_x x\|^2$$

so $\|A_{G(x)}\Lambda_x x\|, \|A_{G(y)}\Lambda_y y\| \neq 0$. Furthermore, note that

$$
\begin{aligned}
\cos\theta_1 &= \frac{\langle \Lambda_x x, A_{G(x)}^\top A_{G(y)}\Lambda_y y \rangle}{\|A_{G(x)}\Lambda_x x\|\|A_{G(y)}\Lambda_y y\|} \\
&= \frac{\langle \Lambda_x x, A_{G(x)}^\top A_{G(y)}\Lambda_y y \rangle}{\sqrt{\langle A_{G(x)}\Lambda_x x, A_{G(x)}\Lambda_x x \rangle \langle A_{G(y)}\Lambda_y y, A_{G(y)}\Lambda_y y \rangle}} \\
&= \frac{\langle \Lambda_x x, \Phi_{G(x),G(y)}\Lambda_y y \rangle + \delta_1}{\sqrt{(\langle \Lambda_x x, \Lambda_x x \rangle + \delta_2)(\langle \Lambda_y y, \Lambda_y y \rangle + \delta_3)}} \\
&= \frac{\langle \Lambda_x x, \Phi_{G(x),G(y)}\Lambda_y y \rangle + \delta_1}{\sqrt{(1 + \delta_2)(1 + \delta_3)}}.
\end{aligned}
$$

Thus if $\epsilon < 1/4L$, we attain

$$
\begin{aligned}
\left| \cos\theta_1 - \langle \Lambda_x x, \Phi_{G(x),G(y)}\Lambda_y y \rangle \right| &\leqslant \left| \frac{\langle \Lambda_x x, \Phi_{G(x),G(y)}\Lambda_y y \rangle + \delta_1}{\sqrt{(1 + \delta_2)(1 + \delta_3)}} - \langle \Lambda_x x, \Phi_{G(x),G(y)}\Lambda_y y \rangle \right| \\
&\leqslant \left| \langle \Lambda_x x, \Phi_{G(x),G(y)}\Lambda_y y \rangle \right| \left| 1 - \frac{1}{\sqrt{(1 + \delta_2)(1 + \delta_3)}} \right| \\
&\quad + \frac{|\delta_1|}{\sqrt{(1 + \delta_2)(1 + \delta_3)}} \\
&\leqslant 2 \left| 1 - \frac{1}{1 - L\epsilon} \right| + \frac{L\epsilon}{1 - L\epsilon} \\
&\leqslant \frac{3L\epsilon}{1 - L\epsilon} \leqslant 4L\epsilon
\end{aligned}
$$

where we used $\|\Phi_{G(x),G(y)}\| \leqslant 2$ in the third inequality. $\qquad\square$

## 5.3 Determining where $h_{x,x_0}$ vanishes

Before proving Proposition 1, we outline how the concentrated gradient $h_{x,x_0}$ was derived. Recall that at points of differentiability, our descent direction is of the following form:

$$v_{x,x_0} = (\Pi_{i=d}^1 W_{i,+,x})^\top A_{G(x)}^\top A_{G(x)}(\Pi_{i=d}^1 W_{i,+,x})x - (\Pi_{i=d}^1 W_{i,+,x})^\top A_{G(x)}^\top A_{G(x_0)}(\Pi_{i=d}^1 W_{i,+,x_0})x_0.$$

The concentration of the first term follows by the RRCP and Lemma 3:

$$(\Pi_{i=d}^1 W_{i,+,x})^\top A_{G(x)}^\top A_{G(x)}(\Pi_{i=d}^1 W_{i,+,x})x \approx (\Pi_{i=d}^1 W_{i,+,x})^\top (\Pi_{i=d}^1 W_{i,+,x})x \approx \frac{1}{2^d}x.$$

For the second term, note that the RRCP gives

$$(\Pi_{i=d}^1 W_{i,+,x})^\top A_{G(x)}^\top A_{G(x_0)}(\Pi_{i=d}^1 W_{i,+,x_0})x_0 \approx (\Pi_{i=d}^1 W_{i,+,x})^\top \Phi_{G(x),G(x_0)}(\Pi_{i=d}^1 W_{i,+,x_0})x_0.$$

Letting $x_d = (\Pi_{i=d}^1 W_{i,+,x})x$ and $x_{0,d} = (\Pi_{i=d}^1 W_{i,+,x_0})x_0$, note that

$$\Phi_{x_d,x_{0,d}} = \frac{\pi - 2\theta_d}{\pi}I_{n_d} + \frac{2\sin\theta_d}{\pi}M_{\hat{x}_d \leftrightarrow \hat{x}_{0,d}}$$

where $\theta_d = \angle(x_d, x_{0,d})$. By Lemma 5 in [20], this angle is well-defined and $\|x_d\|, \|x_{0,d}\| \neq 0$ as long as each $W_i$ satisfies the WDC. Finally, note that the definition of $M_{\hat{x}\leftrightarrow\hat{y}}$ gives

$$M_{\hat{x}_d\leftrightarrow\hat{x}_{0,d}}x_{0,d} = \|x_{0,d}\|M_{\hat{x}_d\leftrightarrow\hat{x}_{0,d}}\hat{x}_{0,d} = \|x_{0,d}\|\hat{x}_d = \frac{\|x_{0,d}\|}{\|x_d\|}x_d.$$

Thus we see that

$$(\Pi_{i=d}^1 W_{i,+,x})^\top \Phi_{x_d,x_{0,d}}(\Pi_{i=d}^1 W_{i,+,x_0})x_0$$
$$= \frac{\pi - 2\theta_d}{\pi}(\Pi_{i=d}^1 W_{i,+,x})^\top(\Pi_{i=d}^1 W_{i,+,x_0})x_0 + \frac{2\sin\theta_d}{\pi}\frac{\|x_{0,d}\|}{\|x_d\|}(\Pi_{i=d}^1 W_{i,+,x})^\top(\Pi_{i=d}^1 W_{i,+,x})x$$
$$\approx \frac{\pi - 2\bar{\theta}_d}{\pi}\tilde{h}_{x,x_0} + \frac{2\sin\bar{\theta}_d}{\pi}\frac{\|x_0\|}{\|x\|}\frac{1}{2^d}x$$

where $\bar{\theta}_d = g^{\circ d}(\angle(x, x_0))$ and the definition of $\tilde{h}_{x,x_0}$ is given in Lemma 3. We recall its definition here for convenience:

$$\tilde{h}_{x,x_0} := \frac{1}{2^d}\left[\left(\prod_{i=0}^{d-1}\frac{\pi - \bar{\theta}_i}{\pi}\right)x_0 + \sum_{i=0}^{d-1}\frac{\sin\bar{\theta}_i}{\pi}\left(\prod_{j=i+1}^{d-1}\frac{\pi - \bar{\theta}_j}{\pi}\right)\frac{\|x_0\|}{\|x\|}x\right].$$

The concentration of the angle $\theta_d$ and norm $\|x_{0,d}\|/\|x_d\|$ are given in Lemma 3. Thus, combining the concentrations of the two terms in $v_{x,x_0}$ gives

$$h_{x,x_0} = \frac{1}{2^d}x - \frac{\pi - 2\bar{\theta}_d}{\pi}\tilde{h}_{x,x_0} - \frac{2\sin\bar{\theta}_d}{\pi}\frac{\|x_0\|}{\|x\|}\frac{1}{2^d}x$$
$$= \frac{1}{2^d}\|x\|\hat{x} - \frac{\|x_0\|}{2^d}\frac{2\sin\bar{\theta}_d}{\pi}\hat{x}$$
$$- \frac{1}{2^d}\left(\frac{\pi - 2\bar{\theta}_d}{\pi}\right)\left[\left(\prod_{i=0}^{d-1}\frac{\pi - \bar{\theta}_i}{\pi}\right)\|x_0\|\hat{x}_0 + \sum_{i=0}^{d-1}\frac{\sin\bar{\theta}_i}{\pi}\left(\prod_{j=i+1}^{d-1}\frac{\pi - \bar{\theta}_j}{\pi}\right)\|x_0\|\hat{x}\right]$$
$$= -\frac{\|x_0\|}{2^d}\left(\frac{\pi - 2\bar{\theta}_d}{\pi}\right)\left(\prod_{i=0}^{d-1}\frac{\pi - \bar{\theta}_i}{\pi}\right)\hat{x}_0$$
$$+ \frac{1}{2^d}\left[\|x\| - \|x_0\|\left(\frac{2\sin\bar{\theta}_d}{\pi} + \left(\frac{\pi - 2\bar{\theta}_d}{\pi}\right)\sum_{i=0}^{d-1}\frac{\sin\bar{\theta}_i}{\pi}\left(\prod_{j=i+1}^{d-1}\frac{\pi - \bar{\theta}_j}{\pi}\right)\right)\right]\hat{x}$$

Now, we establish that the set of all $x$ such that $\|h_{x,x_0}\| \approx 0$, denoted by $S_{\epsilon,x_0}$, is contained in two neighborhoods centered at $x_0$ and a negative multiple $-\rho_d x_0$.

**Proposition 1.** *Suppose* $24\pi d^6\sqrt{\epsilon} \leqslant 1$. *Let*

$$S_{\epsilon,x_0} = \left\{x \in \mathbb{R}^k \setminus \{0\} : \|h_{x,x_0}\| \leqslant \frac{1}{2^d}\epsilon\max(\|x\|, \|x_0\|)\right\}$$

*where* $d \geqslant 2$ *and let*

$$h_{x,x_0} = -\frac{\|x_0\|}{2^d}\left(\frac{\pi - 2\bar{\theta}_d}{\pi}\right)\left(\prod_{i=0}^{d-1}\frac{\pi - \bar{\theta}_i}{\pi}\right)\hat{x}_0$$
$$+ \frac{1}{2^d}\left[\|x\| - \|x_0\|\left(\frac{2\sin\bar{\theta}_d}{\pi} + \left(\frac{\pi - 2\bar{\theta}_d}{\pi}\right)\sum_{i=0}^{d-1}\frac{\sin\bar{\theta}_i}{\pi}\left(\prod_{j=i+1}^{d-1}\frac{\pi - \bar{\theta}_j}{\pi}\right)\right)\right]\hat{x}.$$

*where* $\bar{\theta}_0 = \angle(x, x_0)$ *and* $\bar{\theta}_i = g(\bar{\theta}_{i-1})$. *Define*

$$\rho_d := \frac{2\sin\breve{\theta}_d}{\pi} + \left(\frac{\pi - 2\breve{\theta}_d}{\pi}\right)\sum_{i=0}^{d-1}\frac{\sin\breve{\theta}_i}{\pi}\left(\prod_{j=i+1}^{d-1}\frac{\pi - \breve{\theta}_j}{\pi}\right)$$

*where $\breve\theta_0 = \pi$ and $\breve\theta_i = g(\breve\theta_{i-1})$. If $x \in S_{\epsilon,x_0}$, then either*

$$|\bar\theta_0| \leqslant 2\sqrt{\epsilon} \text{ and } |\|x\| - \|x_0\|| \leqslant 29d\sqrt{\epsilon}\|x_0\|$$

*or*

$$|\bar\theta_0 - \pi| \leqslant 24\pi^2 d^4 \sqrt{\epsilon} \text{ and } |\|x\| - \rho_d\|x_0\|| \leqslant 3517d^8\sqrt{\epsilon}\|x_0\|.$$

*In particular, we have*

$$S_{\epsilon,x_0} \subset \mathcal{B}(x_0, 89d\sqrt{\epsilon}\|x_0\|) \cup \mathcal{B}(-\rho_d x_0, 77422\pi^2 d^{12}\sqrt{\epsilon}\|x_0\|).$$

*Additionally, $\rho_d \to 1$ as $d \to \infty$.*

*Proof.* Without loss of generality, let $x_0 = e_1$ and $\|x_0\| = 1$ where $e_1$ is the first standard basis vector in $\mathbb{R}^k$. We also set $x = \|x\|\left(\cos\bar\theta_0 e_1 + \sin\bar\theta_0 e_2\right)$ where $\bar\theta_0 = \angle(x, x_0)$. Then

$$
h_{x,x_0} = -\frac{1}{2^d}\left(\frac{\pi - 2\bar\theta_d}{\pi}\right)\left(\prod_{i=0}^{d-1}\frac{\pi - \bar\theta_i}{\pi}\right)\hat{x}_0
$$

$$
+ \frac{1}{2^d}\left[\|x\| - \left(\frac{2\sin\bar\theta_d}{\pi} + \left(\frac{\pi - 2\bar\theta_d}{\pi}\right)\sum_{i=0}^{d-1}\frac{\sin\bar\theta_i}{\pi}\left(\prod_{j=i+1}^{d-1}\frac{\pi - \bar\theta_j}{\pi}\right)\right)\right]\hat{x}.
$$

Set

$$
\beta = \left(\frac{\pi - 2\bar\theta_d}{\pi}\right)\left(\prod_{i=0}^{d-1}\frac{\pi - \bar\theta_i}{\pi}\right) \text{ and } \alpha = \frac{2\sin\bar\theta_d}{\pi} + \left(\frac{\pi - 2\bar\theta_d}{\pi}\right)\sum_{i=0}^{d-1}\frac{\sin\bar\theta_i}{\pi}\left(\prod_{j=i+1}^{d-1}\frac{\pi - \bar\theta_j}{\pi}\right)
$$

with $r = \|x\|$ and $M = \max(r, 1)$. Note that we can write

$$
h_{x,x_0} = \frac{1}{2^d}\left(-\beta\hat{x}_0 + (r - \alpha)\hat{x}\right)
$$

Then if $x \in S_{\epsilon,x_0}$, we have that

$$|-\beta + \cos\bar\theta_0(r - \alpha)| \leqslant \epsilon M \tag{25}$$

$$|\sin\bar\theta_0(r - \alpha)| \leqslant \epsilon M. \tag{26}$$

We now tabulate some useful bounds from Lemma 8 in [20]:

$$\bar\theta_i \in [0, \pi/2] \text{ for } i \geqslant 1 \tag{27}$$

$$\bar\theta_i \leqslant \bar\theta_{i-1} \text{ for } i \geqslant 1 \tag{28}$$

$$\left|\prod_{i=0}^{d-1}\frac{\pi - \bar\theta_i}{\pi}\right| \leqslant 1 \tag{29}$$

$$\prod_{i=0}^{d-1}\frac{\pi - \bar\theta_i}{\pi} \geqslant \frac{\pi - \bar\theta_0}{\pi d^3} \tag{30}$$

$$\left|\sum_{i=0}^{d-1}\frac{\sin\bar\theta_i}{\pi}\left(\prod_{j=i+1}^{d-1}\frac{\pi - \bar\theta_j}{\pi}\right)\right| \leqslant \frac{d}{\pi}\sin\bar\theta_0 \tag{31}$$

$$\bar\theta_0 = \pi + O_1(\delta) \implies \bar\theta_i = \breve\theta_i + O_1(i\delta) \tag{32}$$

$$\bar\theta_0 = \pi + O_1(\delta) \implies \left|\prod_{i=0}^{d-1}\frac{\pi - \bar\theta_i}{\pi}\right| \leqslant \frac{\delta}{\pi} \tag{33}$$

$$\left|\frac{\pi - 2\bar\theta_i}{\pi}\right| \leqslant 1 \,\forall\, i \geqslant 1 \tag{34}$$

$$\bar\theta_d \leqslant \cos^{-1}\left(\frac{1}{\pi}\right) \,\forall\, d \geqslant 2 \tag{35}$$

$$\breve\theta_i \leqslant \frac{3\pi}{i+3} \,\forall\, i \geqslant 0. \tag{36}$$

To prove the Proposition, we first show that it is sufficient to only consider the small and large angle case. Then, we show that in the small and large angle case, $x \approx x_0$ and $x \approx -\rho_d x_0$, respectively. We begin by proving that $\max(\|x\|, \|x_0\|) \leqslant 6d$ for any $x \in S_{\epsilon, x_0}$.

**Bound on maximal norm in $S_{\epsilon, x_0}$:** It suffices to show that $r \leqslant 6d$. Suppose $r > 1$ since if $r \leqslant 1$, the result is immediate. Then either $|\sin \bar{\theta}_0| \geqslant 1/\sqrt{2}$ or $|\cos \bar{\theta}_0| \geqslant 1/\sqrt{2}$. If $|\sin \bar{\theta}_0| \geqslant 1/\sqrt{2}$ then (26) gives

$$|r - \alpha| \leqslant \sqrt{2}\epsilon r \implies (1 - \sqrt{2}\epsilon)r \leqslant |\alpha|.$$

But

$$|\alpha| \leqslant \frac{2}{\pi} |\sin \bar{\theta}_d| + \left| \left( \frac{\pi - 2\bar{\theta}_d}{\pi} \right) \sum_{i=0}^{d-1} \frac{\sin \bar{\theta}_i}{\pi} \left( \prod_{j=i+1}^{d-1} \frac{\pi - \bar{\theta}_i}{\pi} \right) \right|$$

$$\leqslant 1 + \frac{d}{\pi}$$

where the second inequality used equations (31) and (34). Thus

$$r \leqslant \frac{1 + \frac{d}{\pi}}{1 - \sqrt{2}\epsilon} \leqslant 2\left(1 + \frac{d}{\pi}\right) \leqslant 2 + d \leqslant 2d$$

provided $\epsilon < 1/4$ and $d \geqslant 2$. If $|\cos \bar{\theta}_0| \geqslant 1/\sqrt{2}$, then (25) gives

$$|r - \alpha| \leqslant \sqrt{2}(\epsilon r + |\beta|) \implies (1 - \sqrt{2}\epsilon)r \leqslant \sqrt{2}|\beta| + \alpha.$$

But by (29),

$$|\beta| = \left| \left( \frac{\pi - 2\bar{\theta}_d}{\pi} \right) \left( \prod_{i=0}^{d-1} \frac{\pi - \bar{\theta}_i}{\pi} \right) \right| \leqslant 1 \text{ since } \bar{\theta}_i \in [0, \pi/2] \; \forall \; i \; \geqslant 1.$$

Hence if $\epsilon < 1/4$,

$$r \leqslant \frac{\sqrt{2} + 2d}{1 - \sqrt{2}\epsilon} \leqslant 2\sqrt{2} + 4d \leqslant \sqrt{2}d + 4d \leqslant 6d.$$

Thus in any case, $r \leqslant 6d \implies M \leqslant 6d$.

We now show that it is sufficient to only consider the small angle case $\bar{\theta}_0 \approx 0$ and the large angle case $\bar{\theta}_0 \approx \pi$.

**Sufficiency:** We have two possible situations:

- $|r - \alpha| \geqslant \sqrt{\epsilon}M$: Then (26) implies
$$|\sin \bar{\theta}_0| \leqslant \sqrt{\epsilon} \implies \bar{\theta}_0 = O_1(2\sqrt{\epsilon}) \text{ or } \pi + O_1(2\sqrt{\epsilon}).$$

- $|r - \alpha| \leqslant \sqrt{\epsilon}M$ : Then (25) implies
$$|\beta| \leqslant 2\sqrt{\epsilon}M.$$

  But note that by (30),
$$\beta = \left( \frac{\pi - 2\bar{\theta}_d}{\pi} \right) \left( \prod_{i=0}^{d-1} \frac{\pi - \bar{\theta}_i}{\pi} \right) \geqslant \frac{(\pi - 2\bar{\theta}_d)(\pi - \bar{\theta}_0)}{d^3 \pi^2}.$$

  In addition, (35) implies
$$|\pi - 2\bar{\theta}_d| \geqslant \left| \pi - 2\cos^{-1}\left( \frac{1}{\pi} \right) \right| \geqslant \frac{1}{2}.$$

  Thus
$$|\beta| \geqslant \frac{|(\pi - 2\bar{\theta}_d)(\pi - \bar{\theta}_0)|}{d^3 \pi^2} \geqslant \frac{|\pi - \bar{\theta}_0|}{2d^3 \pi^2}$$

  which implies
$$|\pi - \bar{\theta}_0| \leqslant 4d^3 \pi^2 \sqrt{\epsilon}M \leqslant 24d^4 \pi^2 \sqrt{\epsilon}.$$

  Thus $\bar{\theta}_0 = \pi + O_1(24d^4 \pi^2 \sqrt{\epsilon})$.

Lastly, we show that in the small angle case, $x \approx x_0$, while in the large angle case, $x \approx -\rho_d x_0$.

**Small Angle Case:** Assume $\overline{\theta}_0 = O_1(2\sqrt{\epsilon})$. Note that since $\overline{\theta}_i \leqslant \overline{\theta}_0 \leqslant 2\sqrt{\epsilon}$ for each $i$, we have that

$$\prod_{i=0}^{d-1} \frac{\pi - \overline{\theta}_i}{\pi} \geqslant \left(1 - \frac{2\sqrt{\epsilon}}{\pi}\right)^d = 1 + O_1\left(\frac{4d\sqrt{\epsilon}}{\pi}\right)$$

provided $2d\sqrt{\epsilon} \leqslant 1/2$. Hence

$$\beta = \left(\frac{\pi - 2\overline{\theta}_d}{\pi}\right)\left(\prod_{i=0}^{d-1} \frac{\pi - \overline{\theta}_i}{\pi}\right)$$

$$\geqslant \left(1 + O_1\left(\frac{4\sqrt{\epsilon}}{\pi}\right)\right)\left(1 + O_1\left(\frac{4d\sqrt{\epsilon}}{\pi}\right)\right)$$

where we used (32) in the second inequality. In addition, $|\sin\overline{\theta}_d| \leqslant |\overline{\theta}_d| \leqslant 2\sqrt{\epsilon}$ and (31) imply that

$$\left|\sum_{i=0}^{d-1} \frac{\sin\overline{\theta}_i}{\pi}\left(\prod_{j=i+1}^{d-1} \frac{\pi - \overline{\theta}_j}{\pi}\right)\right| \leqslant \frac{d}{\pi}|\sin\overline{\theta}_d| \leqslant d\sqrt{\epsilon}.$$

Hence

$$\alpha = \frac{2\sin\overline{\theta}_d}{\pi} + \left(\frac{\pi - 2\overline{\theta}_d}{\pi}\right)\sum_{i=0}^{d-1} \frac{\sin\overline{\theta}_i}{\pi}\left(\prod_{j=i+1}^{d-1} \frac{\pi - \overline{\theta}_j}{\pi}\right)$$

$$= O_1\left(\frac{4\sqrt{\epsilon}}{3\pi}\right) + \left(1 + O_1\left(\frac{4\sqrt{\epsilon}}{\pi}\right)\right)O_1(d\sqrt{\epsilon})$$

$$= O_1\left(\frac{4\sqrt{\epsilon}}{3\pi}\right) + O_1(d\sqrt{\epsilon}) + O_1\left(\frac{4d\epsilon}{\pi}\right)$$

$$= O_1\left(\frac{(4 + 3d\pi + 12d)\sqrt{\epsilon}}{3\pi}\right)$$

Thus since $|-\beta + \cos\overline{\theta}_0(r - \alpha)| \leqslant \epsilon M$ and $M \leqslant 6d$, we attain

$$-\left(1 + O_1\left(\frac{4\sqrt{\epsilon}}{\pi}\right)\right)\left(1 + O_1\left(\frac{4d\sqrt{\epsilon}}{\pi}\right)\right) + (1 + O_1(2\epsilon))\left(r + O_1\left(\frac{(4 + 3d\pi + 12d)\sqrt{\epsilon}}{3\pi}\right)\right)$$

$$= O_1(6d\epsilon).$$

Rearranging, this gives

$$r - 1 = O_1\left(\frac{4d\sqrt{\epsilon}}{\pi} + \frac{4\sqrt{\epsilon}}{\pi} + \frac{16d\epsilon}{\pi} + (2\epsilon + 1)\frac{(4 + 3d\pi + 12d)\sqrt{\epsilon}}{3\pi}\right) + O_1(12d\epsilon) + O_1(6d\epsilon)$$

$$= O_1\left(\frac{(12d + 12 + 48d)\sqrt{\epsilon} + (2\epsilon + 1)(4 + 3\pi d + 12d)\sqrt{\epsilon}}{3\pi} + 18d\sqrt{\epsilon}\right)$$

$$= O_1(29d\sqrt{\epsilon})$$

where we used $\epsilon < 1/2$ in the final equality.

**Large Angle Case:** Assume $\overline{\theta}_0 = \pi + O_1(\delta)$ where $\delta := 24d^4\pi^2\sqrt{\epsilon}$. We first prove that $\alpha$ is close to $\rho_d$. Recall that $\overline{\theta}_d = \breve{\theta}_d + O_1(d\delta)$. Then by the mean value theorem:

$$|\sin\overline{\theta}_d - \sin\breve{\theta}_d| \leqslant |\overline{\theta}_d - \breve{\theta}_d| \leqslant d\delta$$

so $\sin\overline{\theta}_d = \sin\breve{\theta}_d + O_1(d\delta)$. Let

$$\Gamma_d := \sum_{i=0}^{d-1} \frac{\sin\breve{\theta}_i}{\pi}\left(\prod_{j=i+1}^{d-1} \frac{\pi - \breve{\theta}_j}{\pi}\right).$$

Then note that

$$\rho_d = \frac{2\sin\breve\theta_d}{\pi} + \left(\frac{\pi - 2\breve\theta_d}{\pi}\right)\Gamma_d.$$

In [20], it was shown that if $d^2\delta/\pi \leqslant 1$, then $|\Gamma_d| \leqslant d$ and

$$\sum_{i=0}^{d-1}\frac{\sin\overline\theta_i}{\pi}\left(\prod_{j=i+1}^{d-1}\frac{\pi - \overline\theta_j}{\pi}\right) = \Gamma_d + O_1(3d^3\delta).$$

By the condition, $d^2\delta/\pi \leqslant 1$, we require

$$\sqrt\epsilon \leqslant \frac{1}{24\pi d^6}.$$

Thus for sufficiently small $\epsilon$, we have

$$\alpha = \frac{2\sin\overline\theta_d}{\pi} + \left(\frac{\pi - 2\overline\theta_d}{\pi}\right)\sum_{i=0}^{d-1}\frac{\sin\overline\theta_i}{\pi}\left(\prod_{j=i+1}^{d-1}\frac{\pi - \overline\theta_j}{\pi}\right)$$

$$= \frac{2\sin\breve\theta_d}{\pi} + O_1\left(\frac{2d\delta}{\pi}\right) + \left(\frac{\pi - 2\breve\theta_d}{\pi} + O_1\left(\frac{2d\delta}{\pi}\right)\right)(\Gamma_d + O_1(3d^3\delta))$$

$$= \rho_d + O_1\left(\frac{2d\delta}{\pi}\right) + \Gamma_d O_1\left(\frac{2d\delta}{\pi}\right) + \left(\frac{\pi - 2\breve\theta_d}{\pi}\right)O_1\left(3d^3\delta\right) + O_1\left(\frac{6d^4\delta^2}{\pi}\right)$$

$$= \rho_d + O_1\left(\frac{2d\delta}{\pi}\right) + O_1\left(\frac{2d^2\delta}{\pi}\right) + O_1\left(3d^3\delta\right) + O_1\left(\frac{6d^4\delta^2}{\pi}\right)$$

$$= \rho_d + O_1\left(\left(\frac{4\delta}{\pi} + 3\delta + \frac{6\delta^2}{\pi}\right)d^4\right)$$

$$= \rho_d + O_1(7d^4\delta).$$

We now prove $r$ is close to $\rho_d$. Since $x \in S_{\epsilon,x_0}$,

$$|-\beta + \cos\overline\theta_0(r - \alpha)| \leqslant \epsilon M.$$

Also note that $|\beta| \leqslant \delta/\pi$ by 33. Since $\cos\overline\theta_0 = 1 + O_1(\overline\theta_0^2/2)$, we have that

$$O_1(\delta/\pi) + (1 + O_1(\delta^2/2))(r - \rho_d + O_1(7d^4\delta)) = O_1(\epsilon M).$$

Using $r \leqslant 6d$, $\rho_d \leqslant 2d$, and $\delta = 24d^4\pi^2\sqrt\epsilon \leqslant 1$, we get

$$r - \rho_d + O_1\left(\frac{\delta^2}{2}\right)(r - \rho_d) + O_1(7d^4\delta) + O_1\left(\frac{7d^4\delta^3}{2}\right) = O_1(\epsilon M) + O_1\left(\frac{\delta}{\pi}\right)$$

$$\implies r - \rho_d = O_1\left(4d\delta^2 + 7d^4\delta + \frac{7d^4\delta^3}{2} + 6d\epsilon + \frac{\delta}{\pi}\right)$$

$$= O_1\left(6d\epsilon + \delta\left(4d + 7d^4 + \frac{7d^4}{2} + \frac{1}{\pi}\right)\right)$$

$$= O_1\left(\left(6d + 24d^4\pi^2\left(4d + \frac{21d^4}{2} + \frac{1}{\pi}\right)\right)\sqrt\epsilon\right)$$

$$= O_1(3517d^8\sqrt\epsilon).$$

Finally, to complete the proof we use the inequality

$$\|x - x_0\| \leqslant |\|x\| - \|x_0\|| + (\|x_0\| + |\|x\| - \|x_0\||)\overline\theta_0.$$

This inequality states that if a two dimensional point is known to be within $\Delta r$ of magnitude $r$ and an angle $\Delta\theta$ away from 0, then it is at most a Euclidean distance of $\Delta r + (r + \Delta r)\Delta\theta$ away from the point $(r, 0)$ in polar coordinates. Thus for $\overline\theta_0 = O_1(2\sqrt\epsilon)$, we have $r = 1 + O_1(29d\sqrt\epsilon)$ so

$$\|x - x_0\| \leqslant 29d\sqrt\epsilon + (1 + 29d\sqrt\epsilon)2\sqrt\epsilon \leqslant 89d\sqrt\epsilon.$$

Then if $\bar{\theta}_0 = \pi + O_1(24d^4\pi^2\sqrt{\epsilon})$, note that $\angle(x, -\rho_d x_0) = O_1(24d^4\pi^2\sqrt{\epsilon})$ and $r = \rho_d + O_1(3517d^8\sqrt{\epsilon})$ so that

$$
\begin{aligned}
\|x + \rho_d x_0\| &\leqslant 3517d^8\sqrt{\epsilon} + (\rho_d + 3517d^8\sqrt{\epsilon})24d^4\pi^2\sqrt{\epsilon} \\
&\leqslant 3517d^8\sqrt{\epsilon} + (2d + 3517d^8\sqrt{\epsilon})24d^4\pi^2\sqrt{\epsilon} \\
&\leqslant 77422\pi^2 d^{12}\sqrt{\epsilon}.
\end{aligned}
$$

Hence we attain

$$
S_{\epsilon, x_0} \subset \mathcal{B}(x_0, 89d\sqrt{\epsilon}) \cup \mathcal{B}(-\rho_d x_0, 77422\pi^2 d^{12}\sqrt{\epsilon}).
$$

The result that $\rho_d \to 1$ as $d \to \infty$ follows from the following facts: by (36), we have that

$$
\breve{\theta}_d \leqslant \frac{3\pi}{d+3} \; \forall \, d \geqslant 0 \implies \breve{\theta}_d \to 0 \text{ as } d \to \infty.
$$

Thus

$$
\frac{2\sin\breve{\theta}_d}{\pi} \to 0 \text{ as } d \to \infty \text{ since } \breve{\theta}_d \to 0 \text{ as } d \to \infty
$$

and in [20], it was shown that

$$
\sum_{i=0}^{d-1} \frac{\sin\breve{\theta}_i}{\pi} \left( \prod_{j=i+1}^{d-1} \frac{\pi - \breve{\theta}_j}{\pi} \right) \to 1 \text{ as } d \to \infty.
$$

Hence

$$
\left( \frac{\pi - 2\breve{\theta}_d}{\pi} \right) \sum_{i=0}^{d-1} \frac{\sin\breve{\theta}_i}{\pi} \left( \prod_{j=i+1}^{d-1} \frac{\pi - \breve{\theta}_j}{\pi} \right) \to 1 \text{ as } d \to \infty
$$

so $\rho_d \to 1$ as $d \to \infty$. $\qquad\square$

# 6 Appendix B: Gaussian Matrices Satisfy the RRCP

We set out to prove the following:

**Proposition 2.** *Fix $0 < \epsilon < 1$. Let $A \in \mathbb{R}^{m \times n_d}$ have i.i.d. $\mathcal{N}(0, 1/m)$ entries. Then if $m > \tilde{C}_\epsilon dk \log(n_1 n_2 \dots n_d)$, then with probability at least $1 - \tilde{\gamma} m^{4k+1} \exp(-\tilde{c}_\epsilon m)$, $A$ satisfies the RRCP with constant $\epsilon$. Here $\tilde{\gamma}$ is a positive universal constant, $\tilde{c}_\epsilon$ depends on $\epsilon$, and $\tilde{C}_\epsilon$ depends polynomially on $\epsilon^{-1}$.*

To show that Gaussian $A$ satisfies the RRCP, we first establish that for any *fixed* non-zero $z, w \in \mathbb{R}^n$, the inner product $\langle A_z^\top A_w x, y \rangle$ concentrates around its expectation $\langle \Phi_{z,w} x, y \rangle$ for all $x$ and $y$ in a fixed $k$-dimensional subspace of $\mathbb{R}^n$. As we will see by the end of this section, this fixed $k$-dimensional subspace will represent the range of our generative model. We first require a simple technical result that is proven in the subsequent section:

**Proposition 3.** *Fix $z, w \in \mathbb{R}^n \setminus \{0\}$ and $0 < \epsilon < 1$. Let $T$ be a subspace of $\mathbb{R}^n$. If*

$$
\left| \langle A_z^\top A_w x, x \rangle - \langle \Phi_{z,w} x, x \rangle \right| \leqslant \epsilon \|x\|^2 \; \forall \, x \in T \tag{37}
$$

*then*

$$
\left| \langle A_z^\top A_w x, y \rangle - \langle \Phi_{z,w} x, y \rangle \right| \leqslant 3\epsilon \|x\| \|y\| \; \forall \, x, y \in T.
$$

We now require a variation of the Restricted Isometry Property typically proven for Gaussian matrices. In our situation, the matrix $A_z^\top A_w$ concentrates around $\Phi_{z,w} \neq I_n$ for $z \neq w$, so we must prove a generalization which we call the *Restricted Concentration Property* (RCP). First, recall that for any $z, w \in \mathbb{R}^n$, $\mathbb{E}[A_z^\top A_w] = \Phi_{z,w}$. In addition, we have that for any $x \in \mathbb{R}^n$,

$$
\left| \langle A_z^\top A_w x, x \rangle - \langle \Phi_{z,w} x, x \rangle \right| = \frac{1}{m} \left| \sum_{i=1}^{m} Y_i \right|
$$

where

$$Y_i = X_i - \mathbb{E}[X_i] \text{ and } X_i = \text{sgn}(\langle a_i, z\rangle\langle a_i, w\rangle)\langle a_i, x\rangle^2.$$

Here each $a_i$ denotes an unnormalized row of $A$ in which $a_i \sim \mathcal{N}(0, I_n)$. Hence $Y_i$ are independent, centered, subexponential random variables[5]. Thus they satisfy the following large deviation inequality:

**Lemma 5** (Corollary 5.17 in [32]). *Let $Y_1, \ldots, Y_m$ be independent, centered, subexponential random variables. Let $K = \max_{i \in [m]} \|Y_i\|_{\psi_1}$. Then for all $\epsilon > 0$,*

$$\mathbb{P}\left(\frac{1}{m}\left|\sum_{i=1}^m Y_i\right| \geqslant \epsilon\right) \leqslant 2\exp\left[-c\min\left(\frac{\epsilon^2}{K^2}, \frac{\epsilon}{K}\right)m\right]$$

*where $c > 0$ is an absolute constant. Here $\|\cdot\|_{\psi_1}$ is the subexponential norm: $\|X\|_{\psi_1} := \sup_{p \geqslant 1} p^{-1} \left(\mathbb{E}|X|^p\right)^{1/p}$.*

Fix $x \in \mathcal{S}^{n-1}$. Recall that the subexponential norm satisfies

$$\|Y_i\|_{\psi_1} = \|X_i - \mathbb{E}[X_i]\|_{\psi_1} \leqslant 2\|X_i\|_{\psi_1}.$$

Let $Z_i := \langle a_i, x\rangle \sim \mathcal{N}(0, 1)$. Recall that $\|Z_i\|_{\psi_2} \leqslant K_1$ for some absolute constant $K_1$ where $\|\cdot\|_{\psi_2}$ is the sub-gaussian norm. Observe that $\mathbb{E}|X_i|^p \leqslant \mathbb{E}|Z_i^2|^p$. Thus by Lemma 5.14 in [32], we have

$$\|Y_i\|_{\psi_1} \leqslant 2\|X_i\|_{\psi_1} \leqslant 2\|Z_i^2\|_{\psi_1} \leqslant 4\|Z_i\|_{\psi_2}^2 \leqslant 4K_1^2.$$

Thus $K = \max_{i \in [m]} \|Y_i\|_{\psi_1} \leqslant 4K_1^2$ for an absolute constant $K_1$. Defining $K_2 := 4K_1^2$, Lemma 5 guarantees that for any fixed $z, w \in \mathbb{R}^n \setminus \{0\}$ and $\epsilon > 0$,

$$\mathbb{P}\left(|\langle A_z^\top A_w x, x\rangle - \langle \Phi_{z,w} x, x\rangle| \geqslant \epsilon\right) \leqslant 2\exp(-c_0(\epsilon)m) \tag{38}$$

where $c_0(\epsilon) = c\min(\epsilon^2/K_2^2, \epsilon/K_2)$. We are now equipped to proceed with the proof of the RCP.

**Proposition 4** (Variant of Lemma 5.1 in [3]: RCP). *Fix $0 < \epsilon < 1$ and $k < m$. Let $A \in \mathbb{R}^{m \times n}$ have i.i.d. $\mathcal{N}(0, 1/m)$ entries and fix $z, w \in \mathbb{R}^n \setminus \{0\}$. Let $T \subset \mathbb{R}^n$ be a $k$-dimensional subspace. Then if $m \geqslant \tilde{c}k$, we have that with probability exceeding $1 - 2\exp(-c_1 m)$,*

$$|\langle A_z^\top A_w x, x\rangle - \langle \Phi_{z,w} x, x\rangle| \leqslant \epsilon\|x\|^2 \ \forall \ x \in T \tag{39}$$

*and*

$$|\langle A_z^\top A_w x, y\rangle - \langle \Phi_{z,w} x, y\rangle| \leqslant 3\epsilon\|x\|\|y\| \ \forall \ x, y \in T. \tag{40}$$

*Furthermore, let $U = \bigcup_{i=1}^M U_i$ and $V = \bigcup_{j=1}^N V_j$ where $U_i$ and $V_j$ are subspaces of $\mathbb{R}^n$ of dimension at most $k$ for all $i \in [M]$ and $j \in [N]$. Then if $m \geqslant \tilde{c}k$*

$$|\langle A_z^\top A_w u, v\rangle - \langle \Phi_{z,w} u, v\rangle| \leqslant 3\epsilon\|u\|\|v\| \ \forall \ u \in U, \ v \in V, \tag{41}$$

*with probability exceeding $1 - 2MN\exp(-c_1 m)$. Here $c_1$ only depends on $\epsilon$ and $\tilde{c} = \Omega(\epsilon^{-1}\log\epsilon^{-1})$.*

*Proof.* Fix $0 < \epsilon < 1$ and $k < m$. Since $A$ is Gaussian, we may take $T$ to be in the span of the first $k$ standard basis vectors. In addition, assume $\|x\| = 1$ for any $x \in T$. For notational simplicity, set $\Sigma_{z,w} := A_z^\top A_w - \Phi_{z,w}$. Choose a finite set of points $Q_T \subset T$ each with unit norm such that $|Q_T| \leqslant (42/\epsilon)^k$ and for any $x \in T$,

$$\min_{q \in Q_T} \|x - q\| \leqslant \frac{\epsilon}{14}. \tag{42}$$

See [11] for a proof of such a construction. Then we may apply a union bound to (38) for this set of points to attain

$$\mathbb{P}\left(|\langle \Sigma_{z,w} q, q\rangle| \geqslant \frac{\epsilon}{8} \ \forall \ q \in Q_T\right) \leqslant 2\left(\frac{42}{\epsilon}\right)^k \exp\left(-c_0\left(\frac{\epsilon}{8}\right)m\right). \tag{43}$$

Now, define
$$\alpha^* := \inf\left\{\alpha > 0 : |\langle \Sigma_{z,w}x, x\rangle| \leqslant \alpha\|x\|^2 \; \forall \, x \in T\right\}. \tag{44}$$
We want to show that $\alpha^* \leqslant \epsilon$. Fix $x \in T$ with unity norm. Then there exists a $q \in Q_T$ with $\|q\| = 1$ such that $\|x - q\| \leqslant \epsilon/14$. In addition, observe that $x - q \in T$ since $q \in Q_T \subset T$ so by (44),
$$|\langle \Sigma_{z,w}(x - q), x - q\rangle| \leqslant \alpha^*\|x - q\|^2 \leqslant \alpha^*\frac{\epsilon^2}{196}. \tag{45}$$
Now, note that by the definition of $\alpha^*$,
$$|\langle \Sigma_{z,w}x, x\rangle| \leqslant \alpha^* \; \forall \, x \in T.$$
Thus Proposition 3 gives
$$|\langle \Sigma_{z,w}x, y\rangle| \leqslant 3\alpha^* \; \forall \, x, y \in T.$$
Applying this result to $x - q$ and $q$ gives
$$|\langle \Sigma_{z,w}(x - q), q\rangle| \leqslant 3\alpha^*\|x - q\| \leqslant \alpha^*\frac{3\epsilon}{14}. \tag{46}$$
Using $\langle \Sigma_{z,w}x, x\rangle = \langle \Sigma_{z,w}(x - q), x - q\rangle + 2\langle \Sigma_{z,w}x, q\rangle - \langle \Sigma_{z,w}q, q\rangle$ and $\langle \Sigma_{z,w}x, q\rangle = \langle \Sigma_{z,w}(x - q), q\rangle + \langle \Sigma_{z,w}q, q\rangle$, we see that
$$\begin{aligned}
|\langle \Sigma_{z,w}x, x\rangle| &\leqslant |\langle \Sigma_{z,w}(x - q), x - q\rangle| + 2|\langle \Sigma_{z,w}x, q\rangle| + |\langle \Sigma_{z,w}q, q\rangle| \\
&\leqslant |\langle \Sigma_{z,w}(x - q), x - q\rangle| + 2|\langle \Sigma_{z,w}(x - q), q\rangle| + 3|\langle \Sigma_{z,w}q, q\rangle| \\
&\leqslant \alpha^*\frac{\epsilon^2}{196} + \alpha^*\frac{3\epsilon}{7} + \frac{3\epsilon}{8} \\
&= \alpha^*\left(\frac{\epsilon^2}{196} + \frac{3\epsilon}{7}\right) + \frac{3\epsilon}{8}
\end{aligned}$$
where we used (45), (46), and (43) in the second inequality. Note that this bound can be derived for any $x \in T$ because we can always find a $q \in Q_T$ with $\|q\| = 1$ such that $\|x - q\| \leqslant \epsilon/14$. Thus
$$|\langle \Sigma_{z,w}x, x\rangle| \leqslant \alpha^*\left(\frac{\epsilon^2}{196} + \frac{3\epsilon}{7}\right) + \frac{3\epsilon}{8} \; \forall \, x \in T. \tag{47}$$
However, recall that $\alpha^*$ was defined to be the smallest number such that
$$|\langle \Sigma_{z,w}x, x\rangle| \leqslant \alpha^* \; \forall \, x \in T.$$
Hence $\alpha^*$ must be smaller than the right hand side of (47), i.e.
$$\alpha^* \leqslant \alpha^*\left(\frac{\epsilon^2}{196} + \frac{3\epsilon}{7}\right) + \frac{3\epsilon}{8} \implies \alpha^* \leqslant \frac{3\epsilon}{8}\left(\frac{1}{1 - \frac{\epsilon^2}{196} - \frac{3\epsilon}{7}}\right) \leqslant \epsilon$$
since $0 < \epsilon < 1$. Hence we conclude that with probability exceeding $1 - 2(42/\epsilon)^k \exp(-c_0(\epsilon/8)m)$,
$$|\langle \Sigma_{z,w}x, x\rangle| \leqslant \epsilon\|x\|^2 \; \forall \, x \in T$$
i.e.
$$|\langle A_z^\top A_w x, x\rangle - \langle \Phi_{z,w}x, x\rangle| \leqslant \epsilon\|x\|^2 \; \forall \, x \in T.$$
The probability bound in the proposition can be shown by noting that
$$1 - 2(42/\epsilon)^k \exp(-c_0(\epsilon/8)m) = 1 - 2\exp\left(-c_0(\epsilon/8)m + k\log\left(\frac{42}{\epsilon}\right)\right).$$
Thus if
$$\frac{2}{c_0(\epsilon/8)}\log\left(\frac{42}{\epsilon}\right)k \leqslant \tilde{c}k \leqslant m$$
where $\tilde{c} = \Omega(\epsilon^{-1}\log\epsilon^{-1})$, we have that the result holds with probability exceeding
$$1 - 2\exp\left(-c_0(\epsilon/8)m + k\log\left(\frac{42}{\epsilon}\right)\right) \geqslant 1 - 2\exp(-c_1 m)$$
where $c_1 = c_0(\epsilon/8)/2$. Applying Proposition 3 to our result gives (41) with the same probability. The extension to the union of subspaces follows by applying (41) to all subspaces of the form $\text{span}(U_i, V_j)$ and using a union bound.

$$\square$$

Now, this result establishes the concentration of $\langle A_z^\top A_w x, y \rangle$ around $\langle \Phi_{z,w} x, y \rangle$ for $x$ and $y$ in a fixed $k$-dimensional subspace for *fixed* $z, w \in \mathbb{R}^n \setminus \{0\}$. However, in reality, we are interested in showing that this concentration holds for all $z$ and $w$ in the range of our generative model. Hence we require an extension of the RCP, which holds uniformly for all $z$ and $w$ in (possibly) different $k$-dimensional subspaces. We will refer to this result as the Uniform RCP. The proof of this result uses an interesting fact from 1-bit compressed sensing which establishes that if a sufficient number of random hyperplanes cut the unit sphere, the diameter of each tesselation is small with high probability [30]. We state the theorem here for convenience:

**Theorem 4** (Theorem 2.1 in [30]). *Let $n, m, s > 0$ and set $\delta = C_1 \left( \frac{s}{m} \log(2n/s) \right)^{1/5}$. Let $a_i \in \mathbb{R}^n$ have i.i.d. $\mathcal{N}(0,1)$ entries for $i \in [m]$. Then with probability at least $1 - C_2 \exp(-c\delta m)$, the following holds uniformly for all $x, \tilde{x} \in \mathbb{R}^n$ that satisfy $\|x\|_2 = \|\tilde{x}\|_2 = 1$, $\|x\|_1 \leqslant \sqrt{s}$, and $\|\tilde{x}\|_1 \leqslant \sqrt{s}$ for $s \leqslant n$:*

$$\langle a_i, \tilde{x} \rangle \langle a_i, x \rangle \geqslant 0, \ i \in [m] \implies \|\tilde{x} - x\|_2 \leqslant \delta. \tag{48}$$

*Here $C_1, C_2, c$ are positive universal constants.*

We will use this result to prove the following: given a sufficient number of random hyperplanes and a $k$-dimensional subspace $Z$, there exists a finite set of points $Z_0$ such that any point in $Z$ can be closely approximated by a point in $Z_0$ with high probability.

**Lemma 6.** *Fix $0 < \epsilon < 1$. Let $A \in \mathbb{R}^{m \times n}$ have i.i.d. $\mathcal{N}(0, 1/m)$ entries with rows $\{a_\ell\}_{\ell=1}^m$. Let $Z \subset \mathbb{R}^n$ be a $k$-dimensional subspace. Then if $m \geqslant c_\epsilon k$, there exists a set of points*

$$Z_0 := \left\{ z_i \in Z : \|z_i\| = 1 \text{ and } a_\ell^\top z_i \neq 0 \ \forall \ \ell \in [m], \ i \in I \right\} \tag{49}$$

*where $I$ is a finite index set such that the following event holds with probability exceeding $1 - C_2 \exp(-c\epsilon m)$:*

$$E_{Z,A} := \left\{ |I| \leqslant 10m^{2k} \text{ and } \forall \ z \in Z \text{ s.t. } \|z\| = 1, \ \exists \ z_i \in Z_0 \text{ s.t. } \|z - z_i\| \leqslant \epsilon \right\}. \tag{50}$$

*Here $C_2$ and $c$ are positive absolute constants and $c_\epsilon$ depends polynomially on $\epsilon^{-1}$.*

*Proof of Lemma 6.* By the rotational invariance of the Gaussian distribution, we may take $Z$ to be in the span of the first $k$ standard basis vectors. We may further without loss of generality assume $A \in \mathbb{R}^{m \times k}$. Define $Z_0$ and $E_{Z,A}$ as in (49) and (50). We will evoke the following lemma which establishes that the unit sphere of $Z$ is partitioned into at most $10m^{2k}$ regions by the rows $\{a_\ell\}_{\ell=1}^m$ of $A$ with probability 1:

**Lemma 7.** *Let $V$ be a subspace of $\mathbb{R}^n$. Let $A \in \mathbb{R}^{m \times n}$ have i.i.d. $\mathcal{N}(0, 1/m)$ entries. With probability 1,*

$$|\{\operatorname{diag}(\operatorname{sgn}(Av))A : v \in V\}| \leqslant 10m^{2 \dim V}.$$

Now, choose $\{z_i\}_{i \in I}$ as a set of representative points in the interior of each region partitioned by the rows $\{a_\ell\}_{\ell=1}^m$ of $A$. By Lemma 7, the number of such points is bounded with probability 1: $|I| \leqslant 10m^{2k}$. Then, to use Theorem 4, observe that we can set $n = s = k$ since $A \in \mathbb{R}^{m \times k}$ and $Z$ is in the span of the first $k$ standard basis vectors. Then if $m \geqslant \left( C_1^5 \log(2)/\epsilon^5 \right) k := c_\epsilon k$, we have that the quantity $\delta$ in the theorem is bounded by $\epsilon$:

$$\delta := C_1 \left( \frac{k}{m} \log(2) \right)^{1/5} \leqslant \epsilon$$

so $\mathbb{P}(E_{Z,A}) \geqslant 1 - C_2 \exp(-c\epsilon m)$ for some positive universal constants $c, C_1$, and $C_2$ and $c_\epsilon$ depends polynomially on $\epsilon^{-1}$. $\qquad\square$

We now proceed with the proof of the Uniform RCP.

**Proposition 5** (Uniform RCP). *Fix $0 < \epsilon < 1$ and $k < m$. Let $A \in \mathbb{R}^{m \times n}$ have i.i.d. $\mathcal{N}(0, 1/m)$ entries. Let $Z, W$, and $T$ be fixed $k$-dimensional subspaces of $\mathbb{R}^n$. Then if $m \geqslant 2C_\epsilon k$, then with probability at least $1 - 3\gamma m^{4k+1} \exp(-\tilde{c}_\epsilon m)$, we have*

$$\left| \langle A_z^\top A_w x, y \rangle - \langle \Phi_{z,w} x, y \rangle \right| \leqslant L\epsilon \|x\| \|y\| \ \forall \ x, y \in T, \ z \in Z, \ w \in W \tag{51}$$

where $\gamma$ is a positive universal constant, $\tilde{c}_\epsilon$ depends on $\epsilon$ and $C_\epsilon$ depends polynomially on $\epsilon^{-1}$. Furthermore, let $U = \bigcup_{i=1}^M U_i$ and $V = \bigcup_{j=1}^N V_j$ where $U_i$ and $V_j$ are subspaces of $\mathbb{R}^n$ of dimension at most $k$ for all $i \in [M]$ and $j \in [N]$. Then if $m \geqslant 2C_\epsilon k$,

$$\left| \langle A_z^\top A_w u, v \rangle - \langle \Phi_{z,w} u, v \rangle \right| \leqslant L\epsilon \|u\| \|v\| \ \forall \ u \in U, \ v \in V, \ z \in Z, \ w \in W \qquad (52)$$

with probability exceeding $1 - 3MN\gamma m^{4k+1} \exp(-\tilde{c}_\epsilon m)$. Here $L$ is a positive universal constant.

*Proof.* Define $Z_0$ and $E_{Z,A}$ as in (49) and (50). One can define the analogous set

$$W_0 := \left\{ w_j \in W : \|w_j\| = 1 \text{ and } a_\ell^\top w_j \neq 0 \ \forall \ \ell \in [m], \ j \in J \right\} \qquad (53)$$

for some finite index set $J$, choosing the points in $W_0$ in precisely the same way as in $Z_0$. We also define the analogous event

$$E_{W,A} := \left\{ |J| \leqslant 10m^{2k} \text{ and } \forall \ w \in W \text{ s.t. } \|w\| = 1, \ \exists \ w_j \in W_0 \text{ s.t. } \|w - w_j\| \leqslant \epsilon \right\}. \qquad (54)$$

By Lemma 6, we have that if $m \geqslant c_\epsilon k$, $\mathbb{P}(E_{Z,A}) \geqslant 1 - C_2 \exp(-c\epsilon m)$. The event $E_{W,A}$ holds with the same probability so we have that if $m \geqslant c_\epsilon k$,

$$\mathbb{P}(E_{Z,A} \cap E_{W,A}) \geqslant 1 - 2C_2 \exp(-c\epsilon m)$$

For the remainder of this proof, we work on the event $E_{Z,A} \cap E_{W,A}$. Fix $z \in Z$ and $w \in W$. Define the following set:

$$\Omega_{z,w} := \left\{ \ell \in [m] : a_\ell^\top z = 0 \text{ or } a_\ell^\top w = 0 \right\}.$$

Note that since $Z$ and $W$ are $k$-dimensional and any subset of $k$ rows of $A$ are linearly independent with probability 1, at most $k$ entries of either $Az$ or $Aw$ are zero.[6] Hence $|\Omega_{z,w}| \leqslant 2k$. Furthermore, observe that

$$
\begin{aligned}
A_z^\top A_w &= \sum_{\ell=1}^m \mathrm{sgn}(\langle a_\ell, z\rangle \langle a_\ell, w\rangle) a_\ell a_\ell^\top \\
&= \sum_{\ell \in \Omega_{z,w}} \mathrm{sgn}(\langle a_\ell, z\rangle \langle a_\ell, w\rangle) a_\ell a_\ell^\top + \sum_{\ell \in \Omega_{z,w}^c} \mathrm{sgn}(\langle a_\ell, z\rangle \langle a_\ell, w\rangle) a_\ell a_\ell^\top \\
&= \sum_{\ell \in \Omega_{z,w}^c} \mathrm{sgn}(\langle a_\ell, z\rangle \langle a_\ell, w\rangle) a_\ell a_\ell^\top
\end{aligned}
$$

by the definition of $\Omega_{z,w}$. However, on the event $E_{Z,A} \cap E_{W,A}$, there exists a $z_i \in Z_0$ and $w_j \in W_0$ for some $i \in I$ and $j \in J$ such that for all $\ell \in \Omega_{z,w}^c$,

$$\mathrm{sgn}(\langle a_\ell, z\rangle \langle a_\ell, w\rangle) = \mathrm{sgn}(\langle a_\ell, z_i\rangle \langle a_\ell, w_j\rangle)$$

i.e. $z$ and $z_i$ (likewise $w$ and $w_j$) lie on the same side and interior of each hyperplane for which $z$ (or $w$) is not orthogonal to. Hence we have

$$A_z^\top A_w = \sum_{\ell \in \Omega_{z,w}^c} \mathrm{sgn}(\langle a_\ell, z\rangle \langle a_\ell, w\rangle) a_\ell a_\ell^\top = \sum_{\ell \in \Omega_{z,w}^c} \mathrm{sgn}(\langle a_\ell, z_i\rangle \langle a_\ell, w_j\rangle) a_\ell a_\ell^\top := \tilde{A}_{z_i}^\top \tilde{A}_{w_j}.$$

We now use the following lemma which says that if $|\Omega_{z,w}| \leqslant 2k$ total rows of $A_{z_i}$ and $A_{w_j}$ are deleted, we can still establish the RCP:

**Lemma 8.** *Fix $0 < \epsilon < 1$ and $k < m$. Suppose that $A \in \mathbb{R}^{m \times n}$ has i.i.d. $\mathcal{N}(0, 1/m)$ entries. Let $T \subset \mathbb{R}^n$ be a $k$-dimensional subspace and define $Z_0$ and $W_0$ as in (49) and (53). Then if $m \geqslant 2\delta_\epsilon^{-1} \tilde{c} k$, the following holds simultaneously for all $\Omega \subset [m]$ satisfying $|\Omega| \leqslant 2k \leqslant \delta_\epsilon m$ with probability at least $1 - \gamma m^{4k+1} \exp\left(-\frac{c_1 m}{4}\right)$:*

$$\left| \left\langle \tilde{A}_{z_i}^\top \tilde{A}_{w_j} x, y \right\rangle - \langle \Phi_{z_i, w_j} x, y \rangle \right| \leqslant 3\epsilon \|x\| \|y\| \ \forall \ x, y \in T, \ \forall \ i \in I, \ j \in J \qquad (55)$$

*where*

$$\tilde{A}_{z_i}^\top \tilde{A}_{w_j} := \sum_{\ell \in \Omega^c} \mathrm{sgn}(\langle a_\ell, z_i\rangle \langle a_\ell, w_j\rangle) a_\ell a_\ell^\top.$$

*Here $\gamma$ is a positive absolute constant, $c_1$ depends on $\epsilon$, $\tilde{c} = \Omega(\epsilon^{-1} \log \epsilon^{-1})$, and $\delta_\epsilon^{-1}$ depends polynomially on $\epsilon^{-1}$.*

*Proof of Lemma 8.* Fix $\Omega \subset [m]$ satisfying $|\Omega| \leqslant 2k$. For $\delta_\epsilon < 1/2$, observe that the assumption $m \geqslant 2\tilde{c}k$ implies that $|\Omega^c| \geqslant m/2 \geqslant \tilde{c}k$. Thus the RCP guarantees that with probability exceeding

$$1 - 2\exp\left(-c_1|\Omega^c|\right) \geqslant 1 - 2\exp\left(-\frac{c_1 m}{2}\right)$$

we have that the following holds for fixed $z_i \in Z_0$ and $w_j \in W_0$:

$$\left|\left\langle \tilde{A}_{z_i}^\top \tilde{A}_{w_j} x, y \right\rangle - \left\langle \Phi_{z_i, w_j} x, y \right\rangle\right| \leqslant 3\epsilon \|x\| \|y\| \ \forall \ x, y \in T.$$

Furthermore, a union bound over all $\{z_i\}_{i \in I}$ and $\{w_j\}_{j \in J}$ gives

$$\left|\left\langle \tilde{A}_{z_i}^\top \tilde{A}_{w_j} x, y \right\rangle - \left\langle \Phi_{z_i, w_j} x, y \right\rangle\right| \leqslant 3\epsilon \|x\| \|y\| \ \forall \ x, y \in T, \ i \in I, \ j \in J \qquad (56)$$

with probability at least

$$1 - 2|I||J|\exp\left(-\frac{c_1 m}{2}\right) \geqslant 1 - \gamma m^{4k} \exp\left(-\frac{c_1 m}{2}\right)$$

where $\gamma$ is a positive absolute constant and $c_1$ depends on $\epsilon$. The number of subsets of $[m]$ of size $\lfloor \delta_\epsilon m \rfloor$ is

$$\binom{m}{\lfloor \delta_\epsilon m \rfloor} \leqslant \left(\frac{em}{\delta_\epsilon m}\right)^{\delta_\epsilon m} = \left[\left(\frac{e}{\delta_\epsilon}\right)^{\delta_\epsilon}\right]^m$$

We now determine a sufficiently small $\delta_\epsilon$ such that

$$\left(\frac{e}{\delta_\epsilon}\right)^{\delta_\epsilon} \leqslant \exp\left(\frac{c_1}{4}\right) \qquad (57)$$

where $c_1 = c_0(\epsilon/8)/2 = (c/2)\min\left((\epsilon/8)^2/K_2^2, (\epsilon/8)/K_2\right)$ for absolute constants $c$ and $K_2$. Since $0 < \epsilon < 1$, we have that

$$\frac{c_1}{4} \geqslant \frac{c}{8}\min\left(\frac{1}{(8K_2)^2}, \frac{1}{8K_2}\right)\epsilon^2 := R\epsilon^2.$$

Then if $\delta_\epsilon$ satisfies

$$0 \leqslant \exp\left(R\epsilon^2 - \delta_\epsilon\right) - \frac{1}{\delta_\epsilon^{\delta_\epsilon}} \implies \left(\frac{e}{\delta_\epsilon}\right)^{\delta_\epsilon} \leqslant \exp\left(R\epsilon^2\right) \leqslant \exp\left(\frac{c_1}{4}\right).$$

However, note that the function

$$\psi(t) := \exp(t - (t/2)^2) - \frac{1}{(t/2)^{2(t/2)^2}} \geqslant 0 \ \forall \ t > 0.$$

A plot of this function is given in Figure 5. Thus $\psi(R\epsilon^2) \geqslant 0$ so if we take $\delta_\epsilon := (R\epsilon^2/2)^2$, we have that (57) holds.

Defining $\delta_\epsilon$ in this way we have that

$$\binom{m}{\lfloor \delta_\epsilon m \rfloor} \leqslant \exp\left(\frac{c_1 m}{4}\right). \qquad (58)$$

Thus, provided $m \geqslant 2\delta_\epsilon^{-1}\tilde{c}k$ and applying a union bound, the result holds for all subsets $\Omega \subset [m]$ satisfying $|\Omega| \leqslant 2k \leqslant \lfloor \delta_\epsilon m \rfloor$ with probability

$$1 - \sum_{\ell=1}^{\lfloor \delta_\epsilon m \rfloor} \binom{m}{\ell}\gamma m^{4k}\exp\left(-\frac{c_1 m}{2}\right) \geqslant 1 - \lfloor \delta_\epsilon m \rfloor \binom{m}{\lfloor \delta_\epsilon m \rfloor}\gamma m^{4k}\exp\left(-\frac{c_1 m}{2}\right)$$

$$\geqslant 1 - \gamma\lfloor \delta_\epsilon m \rfloor m^{4k}\exp\left(-\frac{c_1 m}{2} + \frac{c_1 m}{4}\right)$$

$$\geqslant 1 - \gamma m^{4k+1}\exp\left(-\frac{c_1 m}{4}\right)$$

where we used (58) in the second inequality. $\qquad \square$

Figure 5: Plot of the function $\psi(t) = \exp(t - (t/2)^2) - \frac{1}{(t/2)^{2(t/2)^2}}$.

We return to the proof of Proposition 5. Let $C_\epsilon := \delta_\epsilon^{-1} \max\{c_\epsilon, \tilde{c}\}$. Then if $m \geqslant 2C_\epsilon k \geqslant 2\tilde{c}k$, Lemma 8 and the event $E_{Z,A} \cap E_{W,A}$ holds with probability exceeding

$$\mathbb{P}\left(\text{Lemma } 8 \cap (E_{Z,A} \cap E_{W,A})\right) \geqslant 1 - 2C_2 \exp\left(-c\epsilon m\right) - \gamma m^{4k+1} \exp\left(-\frac{c_1 m}{4}\right)$$

$$\geqslant 1 - 3\gamma m^{4k+1} \exp\left(-\tilde{c}_\epsilon m\right)$$

where $\gamma$ is a positive absolute constant and $\tilde{c}_\epsilon$ depends on $\epsilon$. On this event, we have that for all $z \in Z$ and $w \in W$ with $\|z\| = \|w\| = 1$, there exists a $z_i \in Z_0$ and $w_j \in W_0$ for some $i \in I$ and $j \in J$ with $\|z_i\| = \|w_j\| = 1$ such that for any $x, y \in T$,

$$\left|\langle A_z^\top A_w x, y\rangle - \langle \Phi_{z,w} x, y\rangle\right| = \left|\langle \tilde{A}_{z_i}^\top \tilde{A}_{w_j} x, y\rangle - \langle \Phi_{z,w} x, y\rangle\right|$$

$$\leqslant \left|\langle \tilde{A}_{z_i}^\top \tilde{A}_{w_j} x, y\rangle - \langle \Phi_{z_i,w_j} x, y\rangle\right| + \left|\langle \Phi_{z_i,w_j} x, y\rangle - \langle \Phi_{z,w} x, y\rangle\right|$$

$$\leqslant 3\epsilon\|x\|\|y\| + \frac{88}{\pi}\epsilon\|x\|\|y\|$$

$$:= L\epsilon\|x\|\|y\|$$

where we used (55) and the continuity of $\Phi_{z,w}$ from Lemma 9 in the second inequality. The extension to the union of subspaces follows by applying (51) to all subspaces of the form $\text{span}(U_i, V_j)$ and using a union bound. $\qquad\square$

With the Uniform RCP, we may now prove the RRCP:

**Proposition 6** (Range Restricted Concentration Property (RRCP)). *Fix* $0 < \epsilon < 1$. *Let* $W_i \in \mathbb{R}^{n_i \times n_{i-1}}$ *have i.i.d.* $\mathcal{N}(0, 1/n_i)$ *entries for* $i = 1, \ldots, d$. *Let* $A \in \mathbb{R}^{m \times n_d}$ *have i.i.d.* $\mathcal{N}(0, 1/m)$ *entries independent from* $\{W_i\}$. *Then if* $m > \tilde{C}_\epsilon dk \log(n_1 n_2 \ldots n_d)$, *then with probability at least* $1 - \tilde{\gamma} m^{4k+1} \exp\left(-\frac{\tilde{c}_\epsilon}{2} m\right)$, *we have that for all* $x, y \in \mathbb{R}^k$,

$$\|(\Pi_{i=d}^1 W_{i,+,x})^\top (A_{x_d}^\top A_{y_d} - \Phi_{x_d, y_d})(\Pi_{i=d}^1 W_{i,+,y})\| \leqslant L\epsilon \prod_{i=1}^d \|W_{i,+,x}\|\|W_{i,+,y}\|$$

*where*

$$x_d := (\Pi_{i=d}^1 W_{i,+,x})x \text{ and } y_d := (\Pi_{i=d}^1 W_{i,+,y})y.$$

*Here* $\tilde{\gamma}$ *and* $L$ *are positive universal constants,* $\tilde{c}_\epsilon$ *depends on* $\epsilon$, *and* $\tilde{C}_\epsilon$ *depends polynomially on* $\epsilon^{-1}$.

*Proof.* It suffices to show that for all $x, y, w, v \in \mathcal{S}^{k-1}$,

$$\left| \langle (A_{x_d}^\top A_{y_d} - \Phi_{x_d, y_d})(\Pi_{i=d}^1 W_{i,+,x})w, (\Pi_{i=d}^1 W_{i,+,y})v \rangle \right| \leqslant L\epsilon \prod_{i=1}^d \|W_{i,+,x}\| \|W_{i,+,y}\|. \quad (59)$$

We will use (52) from Proposition 5. We first consider the $d = 2$ layer case for simplicity. Fix $W_1 \in \mathbb{R}^{n_1 \times k}$ and $W_2 \in \mathbb{R}^{n_2 \times n_2}$. It has been shown in Lemma 15 of [20] that there exists an event $E$ over $(W_1, W_2)$ with $\mathbb{P}(E) = 1$ such that

$$|\{W_{1,+,x} : x \neq 0\}| \leqslant 10 n_1^k \text{ and } |\{W_{2,+,x} : x \neq 0\}| \leqslant 10^2 n_2^k n_1^k.$$

Thus on the event $E$, we have that the following holds with probability 1:

$$|\{W_{2,+,x} W_{1,+,x} : x \neq 0\}| \leqslant 10^3 (n_1^2 n_2)^k.$$

Note that $\dim(\text{range}(W_{2,+,x} W_{1,+,x})) \leqslant k$ for all $x \neq 0$. Hence it follows that

$$\{W_{2,+,x} W_{1,+,x} w : x, w \in \mathcal{S}^{k-1}\} \subseteq U = \bigcup_{i=1}^M U_i$$

where $M \leqslant 10^3 (n_1^2 n_2)^k$. By the same logic, we see that

$$\{W_{2,+,y} W_{1,+,y} v : y, v \in \mathcal{S}^{k-1}\} \subseteq V = \bigcup_{j=1}^N V_j$$

where $N \leqslant 10^3 (n_1^2 n_2)^k$. Thus by applying (52) to $Z = \text{range}(W_{2,+,x} W_{1,+,x})$, $W = \text{range}(W_{2,+,y} W_{1,+,y})$, $U$ and $V$, we see that if $m \geqslant 2C_\epsilon k$, the $d = 2$ layer variant of (59) holds for fixed $W_1$ and $W_2$ with probability exceeding

$$1 - 3MN\gamma m^{4k+1} \exp(-\tilde{c}_\epsilon m) \geqslant 1 - 3(10^3)^2 (n_1^2 n_2)^{2k} \gamma m^{4k+1} \exp(-\tilde{c}_\epsilon m).$$

Let $\tilde{\gamma} = 3(10^3)^2 \gamma$. Observe that if $m \geqslant 2\hat{C} C_\epsilon \tilde{c}_\epsilon^{-1} k \log(n_1 n_2) := \tilde{C}_\epsilon k \log(n_1 n_2)$ for some positive absolute constant $\hat{C}$, then

$$1 - 3(10^3)^2 (n_1^2 n_2)^{2k} \gamma m^{4k+1} \exp(-\tilde{c}_\epsilon m) = 1 - \tilde{\gamma} m^{4k+1} \exp\left(-\tilde{c}_\epsilon m + 2k \log(n_1^2 n_2)\right)$$

$$\geqslant 1 - \tilde{\gamma} m^{4k+1} \exp\left(-\frac{\tilde{c}_\epsilon}{2} m\right).$$

Here $\tilde{\gamma}$ and $\hat{C}$ are positive absolute constants, $\tilde{c}_\epsilon$ depends on $\epsilon$, and $\tilde{C}_\epsilon := 2\hat{C} C_\epsilon \tilde{c}_\epsilon^{-1}$ depends polynomially on $\epsilon^{-1}$. Then, for random $(W_1, W_2)$, we have that by the independence of $A$ and $(W_1, W_2)$, the $d = 2$ layer variant of the RRCP holds with the same probability.

The $d$ layer case is shown with precisely the same argument. It has been shown in Lemma 15 of [20] that

$$|\{\Pi_{i=d}^1 W_{i,+,x} : x \neq 0\}| \leqslant 10^{d^2} (n_1^d n_2^{d-1} \ldots n_{d-1}^2 n_d)^k.$$

Hence it follows that $\{(\Pi_{i=d}^1 W_{i,+,x}) w : x, w \in \mathcal{S}^{k-1}\} \subseteq U$ where $U$ is the union of at most $10^{d^2} (n_1^d n_2^{d-1} \ldots n_{d-1}^2 n_d)^k$ subspaces of dimensionality at most $k$. We can similarly conclude $\{(\Pi_{i=d}^1 W_{i,+,y}) v : y, v \in \mathcal{S}^{k-1}\} \subseteq V$ where $V$ is the union of at most $10^{d^2} (n_1^d n_2^{d-1} \ldots n_{d-1}^2 n_d)^k$ subspaces of dimensionality at most $k$. Hence applying (52) from Proposition 5 to $Z = \text{range}(\Pi_{i=d}^1 W_{i,+,x})$, $W = \text{range}(\Pi_{i=d}^1 W_{i,+,y})$, $U$, and $V$ gives (2) with probability at least

$$1 - \gamma m^{4k+1} (10^{d^2})^2 (n_1^d n_2^{d-1} \ldots n_{d-1}^2 n_d)^{2k} \exp(-\tilde{c}_\epsilon m) \geqslant 1 - \tilde{\gamma} m^{4k+1} \exp\left(-\frac{\tilde{c}_\epsilon}{2} m\right)$$

provided $m \geqslant 2\hat{C} C_\epsilon \tilde{c}_\epsilon^{-1} dk \log(n_1 n_2 \ldots n_d) := \tilde{C}_\epsilon dk \log(n_1 n_2 \ldots n_d)$. $\qquad \square$

## 6.1 RRCP Supplementary Results

*Proof of Proposition 3.* Fix $0 < \epsilon < 1$. Suppose (37) holds and fix $x, y \in T$. Without loss of generality, assume $x$ and $y$ are unit normed. We will use the shorthand notation $\Phi = \Phi_{z,w}$. Since $T$ is a subspace, $x - y \in T$ so by (37),

$$\left|\langle A_z^\top A_w(x - y), x - y\rangle - \langle \Phi(x - y), x - y\rangle\right| \leqslant \epsilon\|x - y\|^2$$

or equivalently

$$\langle \Phi(x - y), x - y\rangle - \epsilon\|x - y\|^2 \leqslant \langle A_z^\top A_w(x - y), x - y\rangle \leqslant \langle \Phi(x - y), x - y\rangle + \epsilon\|x - y\|^2. \tag{60}$$

Note that

$$\|x - y\|^2 = 2 - 2\langle x, y\rangle,$$

$$\langle \Phi(x - y), x - y\rangle = \langle \Phi x, x\rangle + \langle \Phi y, y\rangle - 2\langle \Phi x, y\rangle,$$

and

$$\langle A_z^\top A_w(x - y), x - y\rangle = \langle A_z^\top A_w x, x\rangle + \langle A_z^\top A_w y, y\rangle - 2\langle A_z^\top A_w x, y\rangle$$

where we used the fact that $\Phi$ and $A_z^\top A_w$ are symmetric. Rearranging (60) yields

$$2\left(\langle \Phi x, y\rangle - \langle A_z^\top A_w x, y\rangle\right) \leqslant \left(\langle \Phi x, x\rangle - \langle A_z^\top A_w x, x\rangle\right) + \left(\langle \Phi y, y\rangle - \langle A_z^\top A_w y, y\rangle\right) + (2 - 2\langle x, y\rangle)\epsilon.$$

By assumption, the first two terms are bounded from above by $\epsilon$. Thus

$$2\left(\langle \Phi x, y\rangle - \langle A_z^\top A_w x, y\rangle\right) \leqslant 2\epsilon + (2 - 2\langle x, y\rangle)\epsilon$$
$$= 2(2 - \langle x, y\rangle)\epsilon$$
$$\leqslant 6\epsilon$$

so

$$\langle \Phi x, y\rangle - \langle A_z^\top A_w x, y\rangle \leqslant 3\epsilon.$$

The lower bound is identical. Hence

$$\left|\langle \Phi x, y\rangle - \langle A_z^\top A_w x, y\rangle\right| \leqslant 3\epsilon.$$

$\square$

*Proof of Lemma 7.* It suffices to prove the same upperbound for $|\{\text{sgn}(Av) : v \in V\}|$. Let $\ell = \dim V$. By rotational invariance of Gaussians, we may take $V = \text{span}(e_1, \ldots, e_\ell)$ without loss of generality. Without loss of generality, we may let $A$ have dimensions $m \times \ell$ and take $V = \mathbb{R}^\ell$.[7]

We will appeal to a classical result from sphere covering [36]. If $m$ hyperplanes in $\mathbb{R}^\ell$ contain the origin and are such that the normal vectors to any subset of $\ell$ of those hyperplanes are independent, then the complement of the union of these hyperplanes is partitioned into at most

$$2\sum_{i=0}^{\ell-1}\binom{m-1}{i}$$

disjoint regions. Each region uniquely corresponds to a constant value of $\text{sgn}(Av)$ that has all non-zero entries. With probability 1, any subset of $\ell$ rows of $A$ are linearly independent, and thus,

$$|\{\text{sgn}(Av) : v \in \mathbb{R}^\ell, (Av)_i \neq 0 \,\forall\, i\}| \leqslant 2\sum_{i=0}^{\ell-1}\binom{m-1}{i} \leqslant 2\ell\left(\frac{em}{\ell}\right)^\ell \leqslant 10m^\ell$$

where the first inequality uses the fact that $\binom{m}{\ell} \leqslant (em/\ell)^\ell$ and the second inequality uses that $2\ell(e/\ell)^\ell \leqslant 10$ for all $\ell \geqslant 1$.

For arbitrary $v$, at most $\ell$ entries of $Av$ can be zero by linear independence of the rows of $A$. At each $v$, there exists a direction $\tilde{v}$ such that $(A(v + \delta\tilde{v}))_i \neq 0$ for all $i$ and for all $\delta$ sufficiently small. Hence, $\mathrm{sgn}(Av)$ differs from one of $\{\mathrm{sgn}(Av) : v \in \mathbb{R}^\ell, \ (Av)_i \neq 0 \ \forall \ i\}$ by at most $\ell$ entries. Thus,

$$|\{\mathrm{sgn}(Av) : v \in \mathbb{R}^\ell\}| \leqslant \binom{m}{\ell}|\{\mathrm{sgn}(Av) : v \in \mathbb{R}^\ell, \ (Av)_i \neq 0 \ \forall \ i\}| \leqslant m^\ell 10m^\ell = 10m^{2\ell}.$$

$\square$

We now prove the continuity of $\Phi_{z,w}$ for non-zero $z, w \in \mathbb{R}^n$. Recall that

$$\Phi_{z,w} := \frac{\pi - 2\theta_{z,w}}{\pi}I_n + \frac{2\sin\theta_{z,w}}{\pi}M_{\hat{z}\leftrightarrow\hat{w}}$$

where $\theta_{z,w} := \angle(z, w)$ and $M_{z\leftrightarrow w}$ is the matrix that sends $\hat{z} \mapsto e_1$, $\hat{w} \mapsto \cos\theta_{z,w}e_1 + \sin\theta_{z,w}e_2$, and $h \mapsto 0$ for all $h \in \mathrm{span}(\{z, w\}^\perp)$.

**Lemma 9** (Continuity of $\Phi_{z,w}$). *Fix $0 < \epsilon < 1$ and $z, w \in \mathcal{S}^{n-1}$. Then if $\|\tilde{z} - z\| \leqslant \epsilon$ and $\|\tilde{w} - w\| \leqslant \epsilon$ for some $\tilde{z}, \tilde{w} \in \mathcal{S}^{n-1}$, we have*

$$\|\Phi_{\tilde{z},\tilde{w}} - \Phi_{z,w}\| \leqslant \frac{88}{\pi}\epsilon.$$

*Proof of Lemma 9.* In this proof, we will utilize the following three inequalities:

$$|\theta_{x_1,y} - \theta_{x_2,y}| \leqslant |\theta_{x_1,x_2}|, \ \forall \ x_1, x_2, y \in \mathcal{S}^{n-1} \tag{61}$$

$$2\sin(\theta_{x,y}/2) \leqslant \|x - y\|, \ \forall \ x, y \in \mathcal{S}^{n-1} \tag{62}$$

$$\theta/4 \leqslant \sin(\theta/2), \ \forall \ \theta \in [0, \pi]. \tag{63}$$

Observe that

$$\|\Phi_{\tilde{z},\tilde{w}} - \Phi_{z,w}\| \leqslant \frac{2|\theta_{\tilde{z},\tilde{w}} - \theta_{z,w}|}{\pi}\|I_n\| + \left\|\frac{2\sin\theta_{\tilde{z},\tilde{w}}}{\pi}M_{\tilde{z}\leftrightarrow\tilde{w}} - \frac{2\sin\theta_{z,w}}{\pi}M_{z\leftrightarrow w}\right\|.$$

First, observe that by (61), we have that

$$|\theta_{\tilde{z},\tilde{w}} - \theta_{z,w}| \leqslant |\theta_{\tilde{z},\tilde{w}} - \theta_{z,\tilde{w}}| + |\theta_{z,\tilde{w}} - \theta_{z,w}|$$
$$\leqslant |\theta_{\tilde{z},z}| + |\theta_{\tilde{w},w}|.$$

Then, by (62) and (63), we have that

$$|\theta_{\tilde{z},z}| \leqslant 4\sin(\theta_{\tilde{z},z}/2) \leqslant 2\|\tilde{z} - z\| \leqslant 2\epsilon.$$

The same upper bound holds for $|\theta_{\tilde{w},w}|$. Thus we attain

$$|\theta_{\tilde{z},\tilde{w}} - \theta_{z,w}| \leqslant |\theta_{\tilde{z},z}| + |\theta_{\tilde{w},w}| \leqslant 4\epsilon. \tag{64}$$

Let $R$ be a rotation matrix that maps $z \mapsto e_1$ and $w \mapsto \cos\theta_{z,w}e_1 + \sin\theta_{z,w}e_2$. Let $\tilde{R}$ denote the matrix that applies the same rotatation to the system $\tilde{z}$ and $\tilde{w}$. Recall that $M_{z\leftrightarrow w} := R^\top DR$ and $M_{\tilde{z}\leftrightarrow\tilde{w}} := \tilde{R}^\top \tilde{D}\tilde{R}$ where

$$D := \begin{bmatrix} \cos\theta_{z,w} & \sin\theta_{z,w} & 0 \\ \sin\theta_{z,w} & -\cos\theta_{z,w} & 0 \\ 0 & 0 & 0_{k-2} \end{bmatrix} \text{ and } \tilde{D} := \begin{bmatrix} \cos\theta_{\tilde{z},\tilde{w}} & \sin\theta_{\tilde{z},\tilde{w}} & 0 \\ \sin\theta_{\tilde{z},\tilde{w}} & -\cos\theta_{\tilde{z},\tilde{w}} & 0 \\ 0 & 0 & 0_{k-2} \end{bmatrix}.$$

An elementary calculation shows that $D$ has 2 pairs of non-zero eigenvalues and eigenvectors $(\lambda_1, d_1)$ and $(\lambda_2, d_2)$ where

$$\lambda_1 = -1 \text{ and } d_1 = (\cos\theta_{z,w} - 1)e_1 + \sin\theta_{z,w}e_2$$

while

$$\lambda_2 = 1 \text{ and } d_2 = (\cos\theta_{z,w} + 1)e_1 + \sin\theta_{z,w}e_2.$$

Let $D = -d_1 d_1^\top + d_2 d_2^\top$ be the eigenvalue decomposition for $D$. Then by the definition of $M_{z \leftrightarrow w}$,

$$
\begin{aligned}
M_{z \leftrightarrow w} &= R^\top D R \\
&= R^\top \left( -d_1 d_1^\top + d_2 d_2^\top \right) R \\
&= -R^\top d_1 d_1^\top R + R^\top d_2 d_2^\top R \\
&:= -v_1 v_1^\top + v_2 v_2^\top
\end{aligned}
$$

so $v_1 = R^\top d_1$ and $v_2 = R^\top d_2$ are the eigenvectors of $M_{z \leftrightarrow w}$ with corresponding eigenvalues $-1$ and $1$, respectively. Then, recall that $Rz = e_1$ while $Rw = \cos \theta_{z,w} e_1 + \sin \theta_{z,w} e_2$. Thus the eigenvectors $d_1$ and $d_2$ can be written as

$$
d_1 = Rw - Rz \text{ and } d_2 = Rw + Rz.
$$

Thus the eigenvectors of $M_{z \leftrightarrow w}$ are precisely

$$
v_1 = w - z \text{ and } v_2 = w + z.
$$

By the same argument, the eigenvectors of $M_{\tilde{z} \leftrightarrow \tilde{w}}$ are

$$
\tilde{v}_1 = \tilde{w} - \tilde{z} \text{ and } \tilde{v}_2 = \tilde{w} + \tilde{z}
$$

with corresponding eigenvalues $-1$ and $1$, respectively. Hence, we have that

$$
\begin{aligned}
\frac{2 \sin \theta_{z,w}}{\pi} M_{z \leftrightarrow w} &= \frac{2 \sin \theta_{z,w}}{\pi} \left( -v_1 v_1^\top + v_2 v_2^\top \right) \\
&= \frac{2 \sin \theta_{z,w}}{\pi} \left( -(w-z)(w-z)^\top + (w+z)(w+z)^\top \right)
\end{aligned}
$$

and likewise

$$
\frac{2 \sin \theta_{\tilde{z},\tilde{w}}}{\pi} M_{\tilde{z} \leftrightarrow \tilde{w}} = \frac{2 \sin \theta_{\tilde{z},\tilde{w}}}{\pi} \left( -(\tilde{w}-\tilde{z})(\tilde{w}-\tilde{z})^\top + (\tilde{w}+\tilde{z})(\tilde{w}+\tilde{z})^\top \right).
$$

For simplicity of notation, let $h = w - z$, $\tilde{h} = \tilde{w} - \tilde{z}$, $g = w + z$, and $\tilde{g} = \tilde{w} + \tilde{z}$. Then

$$
\begin{aligned}
\left\| \frac{2 \sin \theta_{z,w}}{\pi} M_{z \leftrightarrow w} - \frac{2 \sin \theta_{\tilde{z},\tilde{w}}}{\pi} M_{\tilde{z} \leftrightarrow \tilde{w}} \right\| &= \frac{2}{\pi} \left\| \sin \theta_{z,w} \left( -hh^\top + gg^\top \right) + \sin \theta_{\tilde{z},\tilde{w}} \left( \tilde{h}\tilde{h}^\top - \tilde{g}\tilde{g}^\top \right) \right\| \\
&\leqslant \frac{2}{\pi} \left( \| \sin \theta_{z,w} hh^\top - \sin \theta_{\tilde{z},\tilde{w}} \tilde{h}\tilde{h}^\top \| + \| \sin \theta_{z,w} gg^\top - \sin \theta_{\tilde{z},\tilde{w}} \tilde{g}\tilde{g}^\top \| \right).
\end{aligned}
$$

Note that since $z, w, \tilde{z}, \tilde{w} \in \mathcal{S}^{n-1}$, $\|h\|, \|\tilde{h}\|, \|g\|, \|\tilde{g}\| \leqslant 2$. In addition,

$$
\|h - \tilde{h}\| \leqslant \|z - \tilde{z}\| + \|w - \tilde{w}\| \leqslant 2\epsilon
$$

and (64) implies

$$
|\sin \theta_{z,w} - \sin \theta_{\tilde{z},\tilde{w}}| \leqslant |\theta_{z,w} - \theta_{\tilde{z},\tilde{w}}| \leqslant 4\epsilon.
$$

Hence

$$
\begin{aligned}
\| \sin \theta_{z,w} hh^\top - \sin \theta_{\tilde{z},\tilde{w}} \tilde{h}\tilde{h}^\top \| &\leqslant \| \sin \theta_{z,w} hh^\top - \sin \theta_{z,w} h\tilde{h}^\top \| + \| \sin \theta_{z,w} h\tilde{h}^\top - \sin \theta_{z,w} \tilde{h}\tilde{h}^\top \| \\
&\quad + \| \sin \theta_{z,w} \tilde{h}\tilde{h}^\top - \sin \theta_{\tilde{z},\tilde{w}} \tilde{h}\tilde{h}^\top \| \\
&\leqslant |\sin \theta_{z,w}| \|h\| \|h - \tilde{h}\| + |\sin \theta_{z,w}| \|\tilde{h}\| \|h - \tilde{h}\| + \|\tilde{h}\tilde{h}^\top\| |\sin \theta_{z,w} - \sin \theta_{\tilde{z},\tilde{w}}| \\
&\leqslant 20\epsilon.
\end{aligned}
$$

The same bound holds for $\| \sin \theta_{z,w} gg^\top - \sin \theta_{\tilde{z},\tilde{w}} \tilde{g}\tilde{g}^\top \|$. Hence we attain

$$
\left\| \frac{2 \sin \theta_{z,w}}{\pi} M_{z \leftrightarrow w} - \frac{2 \sin \theta_{\tilde{z},\tilde{w}}}{\pi} M_{\tilde{z} \leftrightarrow \tilde{w}} \right\| \leqslant \frac{80}{\pi} \epsilon. \tag{65}
$$

Combining (64) and (65), we see that

$$
\|\Phi_{\tilde{z},\tilde{w}} - \Phi_{z,w}\| \leqslant \frac{2|\theta_{\tilde{z},\tilde{w}} - \theta_{z,w}|}{\pi} \|I_n\| + \left\| \frac{2 \sin \theta_{\tilde{z},\tilde{w}}}{\pi} M_{\tilde{z} \leftrightarrow \tilde{w}} - \frac{2 \sin \theta_{z,w}}{\pi} M_{z \leftrightarrow w} \right\| \leqslant \frac{88}{\pi} \epsilon.
$$

$\square$

We prove the inequalities used in the above proof:

*Proof of equations (61), (62), and (63).* For (61), we proceed similarly to the proof on page 12 of [12]. Observe that we can write

$$x_1 = \cos\theta_{x_1,y}y + \sin\theta_{x_1,y}y_1^\perp$$

and

$$x_2 = \cos\theta_{x_2,y}y + \sin\theta_{x_2,y}y_2^\perp$$

where $y_1^\perp$ and $y_2^\perp$ are unit vectors that are orthogonal to $y$. Then observe that

$$\langle x_1, x_2\rangle = \langle\cos\theta_{x_1,y}y + \sin\theta_{x_1,y}y_1^\perp, \cos\theta_{x_2,y}y + \sin\theta_{x_2,y}y_2^\perp\rangle$$
$$= \cos\theta_{x_1,y}\cos\theta_{x_2,y} + \sin\theta_{x_1,y}\sin\theta_{x_2,y}\langle y_1^\perp, y_2^\perp\rangle.$$

Since $\theta_{x_1,y}, \theta_{x_2,y} \in [0,\pi]$, we have that $\sin\theta_{x_1,y}\sin\theta_{x_2,y} \geqslant 0$. In addition, $\langle y_1^\perp, y_2^\perp\rangle \leqslant \|y_1^\perp\|\|y_2^\perp\| = 1$ so we attain

$$\langle x_1, x_2\rangle \leqslant \cos\theta_{x_1,y}\cos\theta_{x_2,y} + \sin\theta_{x_1,y}\sin\theta_{x_2,y} = \cos(\theta_{x_1,y} - \theta_{x_2,y})$$

by the trigonometric identity $\cos(\alpha \mp \beta) = \cos\alpha\cos\beta \pm \sin\alpha\sin\beta$. Since the function $\cos^{-1}(\cdot)$ is decreasing on $[-1,1]$, we see that

$$\theta_{x_1,y} - \theta_{x_2,y} \leqslant \cos^{-1}(\langle x_1, x_2\rangle) = \theta_{x_1,x_2}.$$

Similarly, $\theta_{x_2,y} - \theta_{x_1,y} \leqslant \theta_{x_1,x_2}$ so we attain $|\theta_{x_1,y} - \theta_{x_2,y}| \leqslant |\theta_{x_1,x_2}|$.

For (62), observe that

$$\|x - y\|^2 = \|x\|^2 + \|y\|^2 - 2\langle x, y\rangle$$
$$= \|x\|^2 + \|y\|^2 - 2\|x\|\|y\|\cos\theta_{x,y}$$
$$= 2(1 - \cos\theta_{x,y}).$$

Thus, using the half angle formula

$$\sin\frac{\theta}{2} = \text{sgn}\left(2\pi - \theta + 4\pi\left\lfloor\frac{\theta}{4\pi}\right\rfloor\right)\sqrt{\frac{1 - \cos\theta}{2}}$$

we see that

$$\|x - y\| = \sqrt{2(1 - \cos\theta_{x,y})} = 2\sqrt{\frac{1 - \cos\theta_{x,y}}{2}} \geqslant 2\sin\frac{\theta_{x,y}}{2}.$$

For (63), one can note that the function $\psi(\theta) := 4\sin\frac{\theta}{2} - \theta$ is positive for all $\theta \in [0,\pi]$. $\qquad\square$

## Footnotes

[4]This is the precise constant in the upper bound for the RRCP. Please see the proof of Proposition 5 for its derivation.

[5]Recall that if $a \sim \mathcal{N}(0, I_n)$, $\langle a, x\rangle \sim \mathcal{N}(0, \|x\|^2)$. Since any Gaussian random variable is sub-gaussian and any squared sub-gaussian random variable is subexponential, $\langle a, x\rangle^2$ is subexponential. The terms involving $\text{sgn}(\cdot)$ do not effect the tail of $\langle a, x\rangle^2$.

[6]This is shown in the proof of Lemma 7.

[7]This without loss of generality statement can be deduced by noting the following: if $v \in V \subset \mathbb{R}^n$ where $V$ is an $\ell$-dimensional subspace, then $v = Bq$ where $B \in \mathbb{R}^{n \times n}$ is orthogonal and $q \in \text{span}(e_1, \ldots, e_\ell, 0, \ldots, 0)$. Hence $Av = \tilde{A}q$ where $\tilde{A} = AB$ also has i.i.d. Gaussian entries by the rotational invariance of $A$. Hence it suffices to consider $V = \mathbb{R}^\ell$ and $A \in \mathbb{R}^{m \times k}$.