[Reviews · NeurIPS 2018]

Reviewer 1



Summary: The authors consider the problem of phase retrieval with a particular generative prior for the underlying signal. The generative model is a d-layer, fully-connected, feedforward neural network with ReLU activation functions and no bias terms. The authors consider an \ell_2 empirical risk minimization problem, based on this generative model, and show that the corresponding objective function does not have any spurious local minima or saddle points away from neighborhoods of the true solution (module sign changes). Therefore, if a gradient descent algorithm converges then the solution would be close to the true solution. This result is the main contribution of the paper. Assessment: The paper is well written and well organized, and the work is properly placed in the context of prior art. The contribution is sufficiently original and significant enough to warant acceptance. .

Reviewer 2



This paper considers the phase retrieval problem for a structured signal. In particular, the paper focuses on recovering a structured signal from the magnitudes of its linear measurements, where the measurement vectors are distributed according to the Gaussian distribution. This problem under various structural assumptions on the signal such as sparsity has been extensively considered in the literature. However, the existing efficient recovery methods don't recover the sparse signal with information theoretically optimal sample complexity. Motivated by this observations, this paper explores other structural assumptions which still cover/approximate a large number of signals encountered in practice. In particular, the authors assume that the signal of interest is obtained by mapping a low dimensional latent vector to the ambient dimension with the help of a multi-layer neural network, i.e., the signal has a (deep) generative prior. Such a prior has been recently gaining prominence as an effective prior for a wide class of signals appearing in practice. Working with the deep prior, the authors establish that efficient gradient descent algorithm indeed recovers the signal (the associated latent variable to be precise) with optimal sample complexity. The authors show that the classical gradient descent method converges to the original latent vector or a fixed negative multiple of the latent vector. Relying on the special landscape of the optimization problem at hand, the authors also present a simple modification to the gradient descent method to encourage the convergence to the true latent vector. The paper addresses a problem of practical importance and has a significant amount of novelty both in terms of results and analysis. The paper is well written and conveys all the key ideas in a clear manner. The authors also provide adequate simulations to corroborate their theoretical findings. Overall, the reviewer thinks that the paper nicely covers the prior work in the area of phase retrieval. A few minor comments: 1) Doesn't the PhaseMax have O(n) sample complexity for unstructured signals? 2) The authors may also want to point out the work of Candes and Li which shows O(n) sample complexity for PhaseLift when the signal does not have any structure.

Reviewer 3



This paper considers the sparse phase retrieval (SPR) problem with a generative prior. Specifically, the signal of interest y_0 is assumed to be generated as y_0 = G(x_0), where G is a known neural network and x_0 is an underlying low-dimensional generating seed. Given phaseless observation of the form b = |AG(x_0)|, the task is to recover x_0. This paper focuses on a nonsmooth least-squares formulation for the problem min 1/2* || |AG(x)| - |AG(x_0)| ||^2. The paper shows when the weights of the generating neural network are Gaussian random and the activation function is taken as ReLU, with enough number of measurement (linear in the dimension of the seed vector ignoring dependency on other factors), the least-squares objective has a nice'' global landscape. A gradient descent style algorithm is designed to exploit the particular geometry of the objective. The problem considered is novel. Linear measurement models with deep generative priors have been considered in Refs [4][17]. There is no previous work on the miss-phase problem yet. The geometric result is interesting. Particularly, it implies that no spurious local mins exist for the least-squares objective, except for the global min that is desired and a scaled version of the global min. The proposed numerical algorithm seems simple enough to be practical. My major concern is about the technical message the paper is trying to send. To make the point precise, I quote their text below Theorem 1 (Cf. Theorem 2) as follows: "Our main result asserts that the objective function does not have any spurious local minima or saddle points away from neighborhoods of the true solution and a negative multiple of it. Hence if one were to solve (1) via gradient descent and the algorithm converged, the final iterate would be close to the true solution or a negative multiple of it." 1) Theorem 2 shows outside the two small neighborhoods around the global min and the scaled min, there is always a descent direction v in the sense that the one-sided directional directional derivative defined as D_v(f) = lim_{t \to 0^+} [f(x +tv) - f(x)]/t is negative. This is sufficient for ruling out presence of other local min, but not saddle points. To see this, consider the simple one dimensional function f(x, y) = -|x| + y^2. The one-sided directional derivatives at (0, 0) along both (1, 0) and (-1, 0) direction are -1, but (0, 0) is a stationary point and in fact it is a saddle point. So without further justification, the comment that there are no saddle points seems to be misplaced. To examine presence/absence of saddle points, it seems Clarke subdifferential might be needed. 2) In Fig 1, it may be better to adjust the viewpoint so that more details in the valley can be revealed and also the contour plot can be easily seen. I suspect there are saddle points along the valley connecting the two mins. Please help to confirm/disprove this. 3) If there were saddle points, how the gradient descent method described in Algorithm 1, which is first-order in nature, effectively escapes from the saddle point regions? 4) In theory, it is possible the current iterate is a nondifferentiable point. Algorithm 1 does not specify how to handle such cases. Some additional comments below: * Theorems 1 & 2: It is curious to know what happens within the two small neighborhoods not covered by the current results. As discussed in the paper, this might not matter in practice as \eps can be set to be some small constants so that finding any point would be acceptable numerically. However, this is polynomial dependency of the sample complexity on 1/\eps. Are these fundamental reasons for leaving out the small neighborhoods, or is it just a matter of technicality? I suspect some of the ideas in -- Soltanolkotabi, Mahdi. "Structured signal recovery from quadratic measurements: Breaking sample complexity barriers via nonconvex optimization." arXiv preprint arXiv:1702.06175 (2017). might be useful. * Line 88: Under the Gaussian design model, PhaseLift can recover n-d signals with O(n) measurements due to the refined analysis in -- Candès, Emmanuel J., and Xiaodong Li. "Solving quadratic equations via PhaseLift when there are about as many equations as unknowns." Foundations of Computational Mathematics 14.5 (2014): 1017-1026. * Around Eq (4): In view of the superficial similarity of the formulation in Eq (4) and the current problem in Eq (1), and also the similarity in terms of the geometric result. It might be better to comment on the major innovation on the technical side in proving the current results, compared to Ref [17]. * Theorem 2: I guess here d is treated as a universal constant. Then, the discussion of \rho_d when d tends to infinity seems a bit irrelevant. Also both C and c depend on \eps, which in return is constrained by d due to the restriction that K_1 d^8 \eps^(1/4) \le 1. In the sample complexity m, since the dependency on d is explicitly stated, I think it might be more appropriate to account for the hidden dependency of c on d also. * Sec 3.1: It is stated that "Such a direction exists by the piecewise linearity of the generative model G". I think probably this needs to be elaborated and possibly be shown rigorously. * Exp 4.2: What happens when A consists of Fourier measurements, i.e., corresponding to the classic phase retrieval problem that is of concern in practice? Does the generative modeling offer any conceptual and practical advantage in this more practical setting?